# Nanobodies: From Discovery to AI-Driven Design

**DOI:** 10.3390/biology14050547

**Published:** 2025-05-14

**Authors:** Haoran Zhu, Yu Ding

**Affiliations:** 1State Key Laboratory of Genetics and Development of Complex Phenotypes, School of Life Sciences, Fudan University, Shanghai 200433, China; 22110700099@m.fudan.edu.cn; 2Quzhou Fudan Institute, Quzhou 324002, China

**Keywords:** nanobody, display technologies, nanobody humanization, multiepitope nanobodies, artificial intelligence (AI) in nanobody design

## Abstract

Nanobodies, derived from heavy-chain antibodies in camelids (VHHs) and sharks (V_NAR_s), are single-domain antibodies with unique structural and functional advantages, including small size, high stability, and the ability to bind cryptic or conformational epitopes. Their discovery in the 1990s marked a transformative shift in therapeutic and diagnostic applications, driven by advancements in display technologies, structural biology, and engineering strategies. Key milestones include the first FDA-approved nanobody therapies for thrombotic disorders, autoimmune diseases, and cancer, as well as their use in imaging and biosensors. However, challenges such as immunogenicity and affinity optimization have spurred the integration of AI, enabling rapid structure prediction, humanization, and de novo design. AI tools like AlphaFold3 and ProteinMPNN can now be applied to accelerate rational design, while innovations like NestLink and peptide barcoding will enhance library screening and functional profiling.

## 1. Introduction

Nanobodies (Nbs) [1] and shark V_NAR_s [2] represent two classes of single-domain antibodies with distinct evolutionary origins. Nanobodies derived from camelid VHH antibodies exhibit a unique monomeric architecture (12–15 kDa) with enhanced stability and solubility, enabling bacterial/yeast production systems. Their structural plasticity—particularly their elongated CDR3 loop—facilitates binding to conformational epitopes and cryptic antigenic sites inaccessible to conventional antibodies. Parallel to camelid systems, shark-derived V_NAR_s evolved from IgNAR antibodies through an alternative stabilization mechanism involving CDR1–CDR3 disulfide bonds, granting exceptional pH tolerance and hydrophobic epitope recognition.

Following their initial discovery in camelids during the 1990s, nanobody research has evolved through sequential advancements that have reshaped their scientific trajectory. The field first established robust phage display-based isolation protocols, enabling systematic identification of functional nanobodies [3]. This methodological foundation paved the way for detailed structural characterizations that unraveled the molecular mechanisms behind their exceptional epitope recognition diversity. Building on these insights, recent engineering innovations have unlocked multiepitope designs capable of synergistically enhancing neutralization specificity. Collectively, these developments have propelled nanobodies beyond conventional therapeutic and diagnostic roles into sophisticated biotechnological applications.

The single-domain architecture confers structural versatility supporting functional diversification. While nanobodies exploit their extended CDR3 loops for deep antigen penetration, V_NAR_s leverage rigidified paratopes through unique disulfide bridges. Both systems achieve high-affinity binding across molecular weight spectra—from small metabolites (<500 Da) to viral spike proteins. This structural adaptability underpins emerging strategies like bispecific/multivalent constructs and pH-responsive nanodevices.

Despite their remarkable capabilities, traditional nanobody development pipelines encounter persistent limitations rooted in conventional methodologies. A primary constraint lies in the empirical nature of humanization processes [4], which depend on iterative cycles of mutagenesis to achieve therapeutic compatibility. Compounding this challenge, affinity maturation often necessitates costly experimental campaigns to refine binding specificity. Furthermore, structural prediction inaccuracies frequently arise when dealing with unconventional paratope configurations, particularly those involving complex CDR loop interactions.

Artificial intelligence (AI) emerges as a paradigm-changing solution, with AlphaFold3 enabling the de novo prediction of nanobody–antigen complexes [5] and ProteinMPNN optimizing human-compatible frameworks [6]. Machine learning models systematically map mutation landscapes for simultaneous affinity/humanization optimization, bypassing traditional trial-and-error approaches. Crucially, generative AI facilitates the design of multi-epitope nanobodies through the computational simulation of cooperative binding dynamics—a feat experimentally prohibitive through conventional methods. This AI convergence promises to overcome current limitations while unlocking novel functional modalities.

## 2. Tracing the Historical Discovery of Nanobodies: Key Milestones in Nanobody Research

Understanding the history of nanobody discovery and development is crucial for appreciating the advancements that have shaped their current applications. The journey of nanobodies, starting with their identification in the 1990s, provides valuable insights into how early breakthroughs laid the foundation for their diverse uses in therapeutics, diagnostics, and biotechnology today. Recognizing these milestones not only highlights the evolution of nanobody research but also underscores the significance of key innovations—from camelid antibody discovery to AI-driven design—that have made nanobodies a versatile and powerful tool. For ease of reference, Table 1 below chronologically outlines the major milestones that have driven nanobody research over the years.

The historical journey of nanobodies has been marked by a series of important milestones that span from their initial discovery in camels to the sophisticated AI-driven design methods of today. With continued advancements in structural biology, humanization techniques, and computational design, nanobodies are poised to play an increasingly vital role in therapeutic and diagnostic applications across various fields.

## 3. Overview of Libraries and Display Technologies for Nanobody Development

Building on these foundational milestones, the development of advanced library and display technologies has become pivotal in optimizing nanobody discovery and engineering. As highlighted in the historical timeline, early efforts focused on isolating natural nanobodies from camelids, but modern approaches now leverage diverse library strategies to expand their applicability. Nanobody libraries are generated using different strategies, including naive, synthetic, and immune libraries [24].

Naive libraries are constructed from the genetic material of non-immunized organisms, typically using a collection of germline genes or natural repertoires [25,26]. These libraries do not rely on any prior exposure to the target antigen and are intended to represent a broad spectrum of naturally occurring antibody-like fragments. Naive libraries are generated from untouched genetic material, often from sources like camels, llamas, or alpacas, where the HCAbs are extracted from mRNA and subjected to phage display or other display technologies to create diverse repertoires of nanobodies. The advantages of naive libraries are high diversity and natural diversity; these libraries are derived from natural repertoires, and they may contain sequences that have not been artificially engineered, allowing for a more authentic representation of the binding diversity. Thus, naive libraries can cover a wide range of potential antigen-binding sites, allowing for high variability in the initial selection. However, since the animals are not immunized, the binding affinity of the nanobodies initially generated may be relatively low. Selection methods such as affinity maturation or multiple rounds of selection may be necessary to enhance the affinity of nanobodies for their targets. Naive libraries are widely used when exploring novel targets or uncharacterized epitopes, particularly when starting with a broad or unbiased repertoire.

Synthetic libraries are designed using artificially constructed DNA sequences and can be generated to exhibit a much higher degree of diversity than what is typically found in natural repertoires [17,18,27,28]. These libraries are engineered to mimic the diversity of antibodies but are created with designed randomness. Synthetic libraries are generated by introducing diversity into specific regions of the nanobody gene, such as the CDR3 loop, which is responsible for antigen recognition. This is typically performed by randomizing codons at certain positions during PCR amplification or other DNA synthesis techniques. The resulting library can contain millions to billions of nanobody variants, all expressing the same framework but with different antigen-binding properties. Synthetic libraries can be designed to contain an incredibly high degree of diversity at any chosen site, often much higher than that of naive libraries. Also, synthetic libraries can be rapidly generated without the need for immunization, making them a faster alternative to immune libraries. One of the main drawbacks of synthetic libraries is that, due to issues such as codon bias, they often exhibit the greatest bias among the three library types, which can result in reduced expression yields and lower stability during actual production.

Immune libraries are generated by immunizing an animal (usually a camelid) with a target antigen to stimulate an immune response. The mRNA from the animal’s B cells is then extracted, and nanobodies are synthesized from the immune repertoire [24,29]. These libraries contain a mixture of naturally occurring and affinity-matured nanobodies, resulting from the animal’s immune response. As immune libraries are based on an animal’s natural immune response, the resulting nanobodies often exhibit high affinity and specificity for the target antigen, especially after affinity maturation. Immune libraries often provide faster access to high-affinity binders compared to naive libraries, making them ideal for therapeutic applications. The resulting nanobodies are derived from non-human sources, which may pose an immunogenicity risk when used in humans. To mitigate this, humanization is often required for clinical applications. One of the main drawbacks of immune libraries is that, because they require immunization, they may not be effective when targeting complexes or unstable antigens, as these can dissociate in vivo, preventing a successful immune response. Also, the immunization process takes time, and generating a sufficiently large immune response may be slow compared to that in synthetic libraries.

Historically, display technologies have played a crucial role in the development of nanobodies, enabling the identification and optimization of nanobodies for various applications. These technologies have facilitated the creation of large, diverse nanobody libraries and streamlined the process of selecting high-affinity binders. Given the extensive coverage of display technologies in the literature [30,31,32,33,34,35,36,37,38,39,40,41], with numerous reviews and technical articles describing their detailed applications and mechanisms, this article focuses on providing a concise comparison, which is summarized in Table 2.

In parallel with established display technologies, NestLink has emerged as a novel platform offering unique advantages for nanobody discovery and optimization [49]. NestLink integrates in vivo functional screening with high-throughput sequencing, enabling the simultaneous assessment of nanobody binding and expression stability directly within a cellular context. This approach bridges the gap between synthetic library diversity and the biological relevance of immune libraries, particularly for challenging targets such as membrane proteins or transient molecular interactions. By linking genotype to phenotype in mammalian cells, NestLink preserves post-translational modifications and conformational epitopes critical for targeting complex antigens, a limitation of prokaryotic systems like phage or bacterial display.

Complementing these advancements, peptide barcoding offers a high-throughput, one-pot method to evaluate the sequence–function relationships of free nanobodies [50]. By fusing mutagenized nanobody libraries with unique peptide tags, followed by SEC fractionation and targeted proteomics, this approach identifies critical binding residues (e.g., R35, Y37 in anti-GFP nanobody) and distinguishes subtle K_D_ differences (nM to sub-nM). Validated by SPR and structural analyses, peptide barcoding bypasses display-associated artifacts, enabling precise affinity profiling while maintaining solution-phase authenticity.

NestLink and its related technology’s compatibility with synthetic and immune libraries enhances its versatility. For synthetic libraries, it mitigates codon bias by selecting clones with optimal expression in eukaryotic systems, improving downstream production yields. For immune libraries, it bypasses the need for humanization by directly screening for human-compatible frameworks during selection. Additionally, NestLink’s ability to co-display multiple nanobodies on a single cell facilitates the identification of biparatopic or synergistic binders, accelerating the development of multivalent therapeutics. By combining deep library diversity with physiologically relevant screening environments, NestLink complements existing technologies, offering a streamlined pipeline from library construction to therapeutic-grade candidates. Its integration into nanobody research promises to address longstanding challenges in targeting complex epitopes while accelerating translational applications.

## 4. Clarifying the Unique Properties of Nanobodies Through Structural Biology Analysis: Advantages over Conventional Antibodies

Building on the discussion of library strategies and display technologies, understanding the molecular basis of nanobodies’ unique properties is critical to explaining their success in these systems and guiding their optimization. While library technologies enable the generation and screening of vast nanobody candidates, structural biology provides the foundation for interpreting how their molecular features—such as compact size, thermal stability, and antigen-binding flexibility—directly enhance functional performance. Techniques like X-ray crystallography and cryo-electron microscopy have revealed atomic-level details of nanobody–antigen interactions, revealing why these molecules outperform conventional antibodies in key applications. Below, we dissect the structural characteristics of nanobodies that underpin their advantages.

### 4.1. Single-Domain Structure

Traditional antibodies (Figure 1A) are composed of two heavy chains and two light chains, each with multiple domains, such as variable (VH/VL) and constant regions (CH/CL) [51]. In contrast, antibody fragments and single-domain antibodies simplify this architecture: the Fab fragment (Figure 1B) retains antigen-binding capability through its VH and VL domains (PDB: 1MLC) but lacks the Fc region [52], while the scFv (Single-Chain Variable Fragment, Figure 1C) genetically fuses VH and VL via a flexible linker, reducing the size to around 25 kDa while maintaining specificity (PDB: 1DZB) [53]. Evolutionary adaptations further drive structural minimalism: camelid heavy-chain only antibodies (HCAb) (Figure 1D) [1], derived from camelids, lack light chains entirely (PDB: 1ZVH). The isolated nanobody (single-domain VHH, Figure 1E) exemplifies this simplicity, with its 12~15 kDa molecular weight (PDB: 3K1K) enabling extreme stability and tissue penetration [54]. Similarly, shark-derived V_NAR_ (Figure 1F) (PDB: 1T6V) adopt a β-sheet structure to bind conformational epitopes with flexibility [55]. These single-domain antibodies (VHH and V_NAR_) share advantages: their compact size (12~15 kDa) allows access to cryptic epitopes in dense environments (e.g., tumor microenvironments targeting HER2 [56] or PD-L1 [57]), while their stability allows them to withstand harsh conditions. These features underpin their versatility in immunotherapy, diagnostics, and biotechnology applications [58].

### 4.2. Longer CDR3 Loop

The complementarity-determining region 3 (CDR3) of nanobodies is often longer and more flexible than that of traditional antibodies [59]. In conventional antibodies, the CDR3 loop is relatively short, with less variability compared to nanobodies, typically ranging from 9 to 15 amino acids, whereas in nanobodies, the CDR3 loop can range from 5 to 26 amino acids, contributing to their enhanced flexibility and binding capabilities [60]. The extended and flexible CDR3 loop enables nanobodies to bind to concave surfaces, cryptic epitopes, and other hard-to-target regions on antigens. This enhanced flexibility allows nanobodies to adapt to the shape of the antigen and recognize epitopes that might be inaccessible to larger, more rigid antibodies.

### 4.3. Framework Region Hydrophilicity

The framework regions (FRs) of nanobodies, especially in framework 2 (FR2), contain a high proportion of hydrophilic residues, which contribute to increased solubility and stability in aqueous solutions [61]. These hydrophilic framework regions make nanobodies less prone to aggregation and allow for easy expression in bacterial systems (such as *E. coli*) at high yields. This also contributes to their low immunogenicity when used in human applications, as they are naturally more biocompatible than traditional antibodies [62]. Despite their reduced immunogenicity overall, their highly hydrophilic framework might introduce a structural signature that differs from that of human VH domains, potentially requiring further humanization to minimize immune responses in therapeutic applications.

### 4.4. Absence of the Light Chain

Traditional antibodies rely on the pairing of the heavy chain with the light chain to form a functional antigen-binding site (Figure 1A–C). In contrast, heavy-chain-only antibodies and nanobodies are composed solely of a heavy chain variable domain (VHH) (Figure 1D–F), without a light chain [1]. The absence of the light chain not only simplifies the structure but also contributes to the stability and efficiency of nanobodies in therapeutic applications. However, this structural simplicity also results in rapid renal clearance (half-life < 24 h), necessitating formulation strategies like Fc fusion, PEGylation or albumin binding to achieve therapeutic efficacy in chronic diseases [63]. Nanobodies can be engineered into multivalent formats or fused with other proteins or therapeutic agents to enhance their functional activity. This structural simplicity and modularity make nanobodies ideal candidates for designing bispecific or multispecific antibodies, offering higher target flexibility and reduced steric hindrance [64,65,66].

### 4.5. Greater Stability in Harsh Conditions

Nanobodies exhibit exceptional thermal stability, resistance to denaturation, and can maintain their functional activity across a broad range of pH and temperature conditions [59]. This is due to their compact structure and the rigidity of their framework regions. The ability to remain stable in extreme conditions makes nanobodies ideal for storage, transport, and use in point-of-care diagnostics or biotechnological applications, where more conventional antibodies may require refrigeration or specific conditions to maintain their stability. However, exceptions exist: certain nanobodies may aggregate at high concentrations, despite the general stability of nanobodies as a whole [60].

### 4.6. Efficient Production and Purification

Nanobodies can be efficiently produced in prokaryotic systems, such as *E. coli*, without the need for complex eukaryotic expression systems [33]. Their small size and simple structure enable easy expression and purification, making them ideal for large-scale production. This ability to rapidly produce nanobodies at a low cost and with high yields has made them valuable tools in industrial applications such as biosensors, diagnostics, and therapeutics. Moreover, the easy scalability of production allows for large quantities to be manufactured, facilitating their use in clinical settings. In the research laboratory, this efficiency significantly lowers the barrier to entry for experimental studies, enabling a wider range of researchers, including those with limited resources, to conduct studies on nanobodies. The reduced costs and simplicity in production also make them accessible for high-throughput screening and target identification, democratizing research and fostering innovation across various fields. While nanobodies can be expressed in cost-effective prokaryotic systems (e.g., *E. coli*) or yeast, reducing small-scale production costs, their large-scale manufacturing cost advantage remains uncertain due to the limited industrial adoption in this case compared to IgG. While nanobodies offer production advantages, rabbit antibodies often exhibit higher sensitivity in diagnostic formats due to their greater structural diversity and antigen-binding flexibility [67].

### 4.7. Binding to Concave Surfaces

Nanobodies are especially adept at recognizing concave surfaces on antigens, where conventional antibodies might struggle. These concave regions are often deeply recessed and may be difficult for larger antibodies to access due to their bulkier structure. The flexibility and small size of nanobodies, however, enable the CDR3 loop to extend and adapt to the shape of these recessed regions [68]. A striking example is the EgA1 and 9G8 nanobodies targeting the epidermal growth factor receptor (EGFR) (Figure 2A). Structural studies reveal that these nanobodies bind to a deep cleft at the domain II/III junction of EGFR—a region inaccessible to conventional antibodies like matuzumab due to steric constraints. The nanobodies’ convex paratope, formed by elongated CDR3 loops, penetrates the groove, locking EGFR in its inactive tethered conformation through interactions with residues R310 and E376. This mechanism sterically blocks ligand-induced conformational changes required for receptor activation. In contrast, matuzumab binds a flatter epitope and cannot fully constrain EGFR dynamics. The unique topology of nanobodies enables precise targeting of cryptic concave epitopes critical for allosteric regulation [69].

### 4.8. Binding to Cryptic Epitopes

Nanobodies are particularly useful for binding to cryptic epitopes, which are hidden or inaccessible epitopes on antigens. Nanobodies can bind to the active site cleft of enzymes, where conventional antibodies often fail to penetrate. In a study with hen egg white lysozyme, VHHs isolated from immunized dromedaries predominantly bound to the active site cleft, with their long H3 loops forming convex paratopes that interacted with the enzyme’s catalytic residues [70]. This contrasts conventional antibodies, which typically bind to planar epitopes outside the active site [70]. Another example involves nanobodies selected from immunized llamas that bind to the enzymatic and binding components of Clostridium difficile toxin (CDT). These nanobodies effectively neutralized the cytotoxicity of the binary toxin by blocking the ADP-ribosylation of actin, likely through their long CDR3 loops extending into the NAD-binding cleft of CDTa [71]. In COVID-19-related research, broadly neutralizing bispecific nanobodies (e.g., n3113v and n3130v) have been engineered to target two distinct conserved epitopes on the viral spike protein [72]. These nanobodies efficiently neutralize diverse variants, including Omicron, by n3130v binding to cryptic sites such as the trimer interior (Figure 2B). Notably, the binding sites of n3130v remain unmutated in highly prevalent variants, conferring robust cross-reactivity. This strategy leverages nanobodies’ ability to access buried epitopes, a critical advantage in countering viral immune evasion. Detailed mechanisms will be discussed in subsequent sections.

### 4.9. Binding to Other Hard-to-Target Regions

Beyond concave surfaces and cryptic epitopes, nanobodies are capable of binding to other hard-to-target regions on antigens that larger antibodies cannot access. A prominent example is their application in studying G protein-coupled receptors (GPCRs), dynamic membrane proteins central to cellular signaling [73,74]. Nanobodies excel at stabilizing transient conformational states of GPCRs, such as active or inactive forms, enabling structural insights into their activation mechanisms. For instance, nanobodies like Nb80 lock the β2-adrenergic receptor (β2AR) in its agonist-bound active state by binding intracellular domains critical for G protein coupling (Figure 2C). This approach facilitated the first high-resolution structures of active β2AR and revealed conserved activation features [75]. Similarly, nanobodies targeting the μ-opioid receptor (Nb39) and M2 muscarinic receptor (Nb9-8) stabilize distinct ligand-bound states, elucidating receptor-specific activation pathways [76,77]. Nanobodies also stabilize GPCR complexes with signaling partners, such as the β2AR–Gs complex, by bridging interfaces to prevent dissociation. For cryo-EM studies, engineered “megabodies” and “legobodies” overcome size limitations, resolving GPCR–G protein complexes or other membrane protein complexes at near-atomic resolutions [78,79]. Additionally, conformation-specific nanobodies enable drug discovery by screening compounds against locked receptor states, enhancing selectivity for therapeutic agents. Their ability to access dynamic or sterically constrained regions positions them as indispensable tools for investigating GPCR pharmacology and allostery.

Building on this versatility in targeting dynamic proteins, nanobodies have also emerged as critical tools for studying small GTPases, such as RhoA, which function as molecular switches by cycling between inactive GDP-bound and active GTP-bound states to regulate actin cytoskeleton dynamics. The recently discovered nanobody Rh57 selectively binds to the active GTP-bound RhoA without causing cytotoxic effects, making it a promising biosensor [28,80]. Structural analysis of the RhoA–Rh57 complex revealed that the interaction is primarily mediated by hydrogen bonds, salt bridges, aromatic interactions, and hydrophobic contacts, involving Rh57’s CDR3 and non-hypervariable loops and RhoA’s SWI switch regions [81]. Importantly, Rh57 does not interfere with downstream signaling, as its binding site does not overlap with key effectors such as PRK1. Unlike the previously identified nanobody Rh12, which disrupted actin fibers and affected cell viability, Rh57 offers a non-invasive alternative for the intracellular monitoring of RhoA activation. Additionally, the development of a BRET-based biosensor using Rh57 enables the real-time tracking of RhoA activation dynamics and provides a valuable tool for screening potential modulators of the RhoA subfamily. These findings offer critical insights for the further optimization and development of nanobody-based biosensors targeting small GTPases.

### 4.10. Comparisons with Alternative Technologies

While nanobodies excel in targeting cryptic epitopes and enabling modular design, competing technologies like DARPins and Affibodies offer complementary strengths. DARPins’ rigid ankyrin repeats provide superior stability for industrial applications [82], whereas Affibodies’ modular architecture facilitates probe conjugation for imaging [83]. These trade-offs highlight the need for technology selection based on target requirements.

## 5. Comparison of Structural Differences Among Camelid-Derived, Human-Derived, and Shark-Derived Nanobodies

Nanobodies from camelids (camels, llamas, and alpacas) and V_NAR_ from sharks, while functionally similar, differ significantly in their structure, which impacts their binding properties, stability, and potential applications. We summarize the structural differences between camelid, human, and shark-derived nanobodies in Table 3 and Figure 3, providing a comparative overview of their unique features, advantages, and limitations.

Camelid-derived nanobodies (VHHs) consist of a single variable domain that functions as the antigen-binding site [1]. Structurally, they are composed of four framework regions (FR1–FR4) and three complementarity-determining regions (CDR1–CDR3) arranged from the N- to C-terminus (Figure 3A). Beyond the canonical CDRs, studies have identified contributions to antigen binding from residues within FR3, which are sometimes designated as a putative CDR4. Additionally, a conserved disulfide bond between residues preceding CDR1 and CDR3 stabilizes the overall architecture. The CDR3 loop of VHHs is typically longer and more flexible than that of conventional antibodies, enabling the effective recognition of cryptic epitopes or concave antigen surfaces (Figure 3B) [84]. Notably, the framework regions of camelid nanobodies exhibit high hydrophilicity, which enhances solubility and minimizes aggregation. This property is significantly influenced by residues in the FR2 region, including the highly conserved F37, E44, R45, and G47 (Kabat numbering, highlighted in Figure 3B) [85], which collectively stabilize the domain and optimize aqueous solubility. Evolutionarily, camelid VHHs inherently compensate for the absence of a light chain through structural adaptations in the framework regions, particularly FR2. These conserved residues and the unique disulfide bonding pattern confer robust stability and high-affinity binding to protein antigens without relying on light-chain interactions.

Human-derived nanobodies are engineered from the variable heavy (VH) domain of conventional antibodies to mimic the properties of camelid nanobodies while being fully human, reducing immunogenicity (Figure 3C). However, their CDR3 loops are shorter and less flexible, limiting their ability to access certain recessed epitopes. They often require engineering to enhance stability, solubility, and binding affinity, as they lack the natural hydrophilic mutations found in camelid nanobodies. Human-derived nanobodies rely heavily on rational design and directed evolution for functional optimization.

Shark-derived nanobodies (V_NAR_s), unique to cartilaginous fish such as sharks, represent the smallest known antibody fragments and exhibit a highly compact structure [86]. Their elongated CDR3 loop enables binding to narrow grooves and cryptic epitopes (Figure 3D). In contrast to camelid VHHs (Figure 3B), V_NAR_s display a more slender architecture, with an additional disulfide bond between CDR1 and CDR3 (characteristic of Type 2 V_NAR_s) that stabilizes their tertiary structure [87]. Unlike camelid and human-derived nanobodies, V_NAR_s adopt a distinct nomenclature system where the truncated CDR2 region is designated as hypervariable region 2 (HV2). Importantly, the HV4 region—a unique structural element located between framework regions 3 and 4—has been implicated in antigen binding in certain cases. V_NAR_s are naturally highly stable, capable of functioning in extreme conditions such as high salinity or pH extremes. They are structurally diverse and are particularly well suited for binding non-protein antigens, including carbohydrates and lipids. However, V_NAR_s face certain limitations, particularly in their low sequence homology with human antibodies, making humanization more challenging compared to camelid or human-derived nanobodies. The substantial structural differences can lead to a higher risk of immunogenicity in therapeutic applications unless extensive engineering is performed. Additionally, the unique reliance of V_NAR_s on their CDR3 loop for binding may restrict their adaptability to some target classes, requiring further optimization for complex or conformational epitopes. These challenges highlight the need for advanced design and engineering strategies when developing V_NAR_-based therapeutics.

## 6. Humanization Strategy for Camelid Nanobodies

For camelid nanobodies to be developed for clinical applications, they must undergo humanization to minimize the risk of immunogenicity when introduced into the human body. This process ensures the nanobody is compatible with the human immune system while preserving its functional properties. A key aspect of this humanization involves modifications to the framework 2 (FR2) region, which exhibits four critical amino acid differences between camelid VHHs and human VH domains. These differences are essential for the unique structure and solubility of camelid nanobodies, enabling them to function as heavy-chain-only antibodies without a light chain. The camelid-specific residues and their human counterparts are as follows: Phe/Tyr-37 → Val, Glu-44 → Gly, Arg-45 → Leu, and Gly-47 → Trp (Kabat Database) [4,85,88,89].

Each of the four camelid-specific amino acids serves a distinct functional role. Phenylalanine or tyrosine at position 37, replaced by valine in human VH domains, enhances the hydrophilicity of camelid nanobodies. Glutamic acid at position 44, replaced by glycine in humans, contributes to solubility and structural stability. Arginine at position 45, substituted with leucine, supports antigen binding and overall structural integrity. Finally, glycine at position 47, which is tryptophan in human antibodies, prevents aggregation by increasing hydrophilicity (Figure 3B,C). To humanize camelid nanobodies, these four amino acids in FR2 are substituted with their human equivalents. This requires precise engineering to preserve the nanobody’s binding affinity and structural stability. Computational modeling, such as molecular dynamics simulations, is often employed to validate the impact of these substitutions, ensuring the antigen-binding site remains functional. Following this, the humanized nanobody undergoes expression in suitable systems (e.g., *E. coli* or mammalian cells) and rigorous functional testing, such as binding assays and stability analyses, to confirm its performance. This careful balance between humanization and functional preservation is critical for advancing nanobodies into clinical settings. However, residual immunogenicity risks still persist due to non-human framework residues, as evidenced in clinical trials where anti-nanobody antibodies were still detected in some patients despite humanization efforts [62,90]. Recent advancements in AI have significantly enhanced the humanization process by enabling more accurate prediction of sequence modifications and their functional outcomes. The application of AI-driven approaches to nanobody engineering will be further discussed in this review.

While humanizing camelid nanobodies reduces immunogenicity, a more advanced approach is the direct use of fully human nanobody libraries, which bypasses the need for humanization altogether. Certain isolated human VH domains have been identified with biophysical properties comparable to those of camelid nanobodies [88,91,92]. Structural studies suggest that specific VH framework regions can compensate for the absence of a light chain, enabling the formation of soluble human single-domain antibodies (UdAbs). A notable example is n501, a UdAb targeting the oncofetal antigen 5T4, associated with solid tumors [93]. High-resolution crystal structures reveal that n501 adopts a compact structure nearly identical to that of camelid nanobodies, binding tightly to all eight leucine-rich repeats of 5T4. This UdAb demonstrates remarkable stability, retaining its activity even after 4 weeks of storage at various temperatures. When conjugated with the cytotoxic payload SN38 to form the antibody–drug conjugate (ADC) n501-SN38, the UdAb exhibited superior tumor penetration, faster accumulation at tumor sites, and significantly higher tumor uptake compared to conventional IgG1-based ADCs (e.g., m603-SN38). These properties translated into improved tumor inhibition in preclinical models, showcasing the promise of UdAb-based ADCs as a novel class of antitumor therapeutics with enhanced stability, penetration, and efficacy against solid tumors.

## 7. Nanobodies and Their Unique Binding Mode with Specific Antigens

Nanobodies, leveraging their compact single-domain architecture and hyperflexible CDR3 loops, exhibit unparalleled versatility in molecular recognition across diverse targets—from small molecules (<1000 Da) to cryptic epitopes and non-protein antigens. These antigen-binding fragments evolve unique structural strategies to overcome conventional antibody limitations, including the following: (1) multi-CDR cooperative binding and CDR1-tunneling modes for small-molecule encapsulation; (2) non-canonical disulfide bonds to stabilize compact binding cavities; (3) homodimerization-driven recognition for enhanced avidity; and (4) intrinsic dual-epitope engagement through spatially segregated CDR loops. Such innovations are further augmented by their ability to penetrate hydrophobic pockets and bind carbohydrates or lipids, exemplified by shark-derived V_NAR_s. These structural adaptations not only enable high-affinity interactions with challenging targets—such as transiently exposed toxin epitopes or GTPase activation states—but also minimize immunogenicity risks through natural or engineered frameworks. By integrating crystallographic insights with functional mutagenesis, nanobodies are redefining precision in therapeutic intervention, biosensing, and environmental monitoring, while offering a roadmap for the rational design of next-generation antibody mimics.

### 7.1. CDR1-Tunneling Modes for Small-Molecule Encapsulation

Crystallographic studies of nanobody–small molecule complexes deposited in the PDB reveal two primary binding mechanisms. The first resembles the canonical binding mode of conventional antibodies, where multiple CDRs (CDR1, CDR2, and CDR3) cooperatively engage the antigen. For instance, nanobodies targeting larger molecules such as the azo dyes Reactive Red 6 (RR6, ~800 Da) RR6 [94] (Figure 4A) and Reactive Red 1 RR1 (RR1, ~500 Da) [95] employ this multi-CDR strategy. In these cases, the CDR3 loop, along with contributions from CDR1 and CDR2, forms a binding pocket that accommodates the bulky aromatic structures of the dyes. This mode underscores the adaptability of nanobodies in recognizing diverse epitopes through combinatorial CDR interactions.

The second mechanism, termed the CDR1-tunneling mode, is exclusive to nanobodies and is observed for smaller hydrophobic haptens (<500 Da), including methotrexate (MTX; 454 Da) [96], triclocarban (TCC; 316 Da) [97], and cortisol (362 Da) [98] (Figure 4B). Here, the antigen is embedded within a hydrophobic tunnel formed by the CDR1 loop, with additional contacts mediated by CDR4. This mode maximizes the contact surface area despite the antigen’s small size, ensuring high binding affinity. This mechanism is further exemplified by the anti-quinalphos nanobody Nb-11A, which recognizes O,O-diethyl thiophosphate-containing pesticides (14.3–174.0 nM affinity range) through a CDR1/FR3-mediated tunnel [99]. Structural analyses reveal that ligand orientation and hydrophobicity govern binding affinity differences, while cavity enlargement and hydrogen bond rearrangement under stress conditions explain its stability limitations. Emerging evidence suggests this mechanism may extend to ustilaginoidin A recognition, where preliminary conformational analyses show that one naphtho-γ-pyrone ring of the mycotoxin inserts into a pocket formed by FR1, CDR1, and CDR3 in Nb-B15/Nb-C21, while the second ring remains surface-exposed—potentially enabling dimer formation through inter-nanobody bridging [100]. Final confirmation of ustilaginoidin A’s binding mode awaits crystal structure determination. Notably, this tunneling strategy avoids reliance on the CDR3 loop, reflecting a structural innovation unique to nanobodies.

### 7.2. Non-Canonical Disulfide Bonds to Stabilize Compact Binding Cavities

Intriguingly, the H1-17 nanobody, which binds sulfadimethoxine (SDM; 310 Da), challenges the above classifications with a novel binding paradigm using CDR3 and non-canonical disulfide bonds [101]. Docking and mutagenesis analysis reveals a CDR3-dominated binding mode augmented by a non-canonical disulfide bond (C50-C104). Unlike conventional or tunneling modes, SDM recognition depends almost exclusively on CDR3, with the disulfide bond compensating for the unusually short CDR3 loop. This bond stabilizes a compact binding cavity that precisely accommodates SDM’s sulfonamide core and methoxy group. Mutagenesis studies confirm that the disulfide bond is critical for maintaining the pocket’s geometry, enabling H1-17 to achieve ultra-specificity—exhibiting no cross-reactivity with 41 structural analogs. This mechanism highlights how nanobodies exploit non-classical structural features to overcome limitations in CDR length, broadening their applicability to small, structurally similar molecules.

### 7.3. Homodimerization-Driven Recognition for Enhanced Avidity

Nanobodies further diverge from conventional antibodies through homodimerization-driven antigen recognition. The anti-caffeine VHH exemplifies this strategy, binding caffeine with a 2:1 stoichiometry (Figure 4C). In the absence of caffeine, the VHH forms a stable homodimer resembling the VH-VL heterodimer of traditional antibodies but with a smaller interaction angle and enhanced hydrophobic surface burial [102,103,104]. Caffeine binding stabilizes this dimer, with Tyr34 and Tyr104 forming hydrogen bonds and π-stacking interactions critical for specificity. This dimeric recognition mode is not isolated; anti-picloram VHHs with short CDR3 loops also adopt homodimeric configurations, suggesting a broader paradigm for small-molecule targeting [105]. Such homodimers expand the functional repertoire of nanobodies, enabling applications in chemical-induced dimerization systems and multivalent assays.

As crystallographic and mutagenesis studies continue to unravel these mechanisms, the rational design of nanobodies with tailored specificity and stability will advance their use in precision diagnostics, therapeutics, and environmental monitoring. Future research should explore the generality of homodimer-driven recognition and engineer non-natural disulfide bonds to expand the nanobody toolbox for challenging small-molecule targets.

### 7.4. Intrinsic Dual-Epitope Engagement Through Spatially Segregated CDR Loops

Nanobody SEB-Nb6 represents a rare paradigm wherein a single-domain antibody concurrently engages two non-overlapping epitopes (domains I and II) on adjacent Staphylococcal enterotoxin B (SEB) via distinct CDR loops (CDR1/2/non-hypervariable loop for domain I on one SEB molecule; CDR3/non-hypervariable region for domain II on an adjacent SEB molecule) (Figure 4D) [24,106]. Unlike engineered tandem nanobodies requiring artificial linkers, Nb6’s natural dual-targeting mechanism preserves a minimal molecular weight (~15 kDa), enhancing tissue penetration while avoiding steric constraints. Critically, this dual-epitope engagement exploits SEB’s unique surface topology, which features two spatially adjacent yet structurally independent binding crevices (Domains I and II) on neighboring SEB molecules. By bridging these sites, Nb6 induces the formation of a ternary complex (SEB–nanobody–SEB), triggering a cooperative aggregation effect that disrupts SEB’s toxic oligomerization or receptor binding. Furthermore, multiple Nb6 molecules can cross-link SEB monomers into large supramolecular assemblies. This mechanism—dependent on the antigen’s inherent dimeric or multimeric configuration—is not universally applicable but demonstrates a novel strategy to amplify therapeutic efficacy through antigen self-assembly. Nb6’s unique mode of action highlights the potential of exploiting fortuitous antigen architectures (e.g., adjacent independent epitopes) to design compact, high-avidity therapeutics, though its rarity underscores the need for advanced epitope-mapping technologies to identify analogous targets.

## 8. Nanobodies Binding to Multiple Epitopes of the Same Antigen

Nanobodies exhibit exceptional versatility by binding to multiple epitopes on the same antigen. This ability to simultaneously or sequentially recognize distinct regions of a single antigen offers significant advantages for therapeutic applications, diagnostics, and biosensing. To maintain brevity, we highlight detailed examples involving nanobodies targeting multiple epitopes on well-known antigens such as the COVID-19 spike protein, influenza virus hemagglutinin (HA) protein, GFP, and mCherry.

### 8.1. Nanobodies Targeting Different Epitopes on the COVID-19 Spike Protein

The SARS-CoV-2 spike protein (S protein), a trimeric glycoprotein critical for viral entry, is the primary target for the immune response in COVID-19. Nanobodies have been developed to target multiple epitopes on the spike protein to enhance neutralization and prevent viral entry into human cells [107]. The S protein’s epitopes span the receptor-binding domain (RBD), N-terminal domain (NTD), and conserved regions of the S2 subunit [108]. For instance, RBD-targeting nanobodies such as VHH72, mNb6 and Ty1 block ACE2 receptor binding by directly occluding the interaction interface, achieving potent neutralization [109,110,111]. In contrast, nanobodies like XG2v046 and XGv280 bind to the NTD, allosterically promoting RBD “up” conformation and S1 subunit shedding, thereby destabilizing the prefusion spike trimer and disrupting viral fusion machinery [112]. Targeting the S2 subunit of the SARS-CoV-2 spike protein represents a promising strategy due to its conserved nature, enabling broad neutralization against evolving variants. Current research primarily focuses on conventional monoclonal antibodies (e.g., IgG) targeting S2 epitopes such as the stem helix, fusion peptide (FP), or heptad repeat (HR) regions. These antibodies disrupt viral fusion by stabilizing prefusion conformations or blocking postfusion structural rearrangements. However, S2-specific nanobodies remain underexplored, likely due to challenges in epitope accessibility and discovery methods tailored for traditional antibodies [113].

Koenig et al. engineered multivalent nanobodies, such as biparatopic VE/EV and homotrivalent EEE, by the structurally guided fusion of RBD-targeting VHHs, demonstrating potent blockading of SARS-CoV-2 infection and robust suppression of viral escape through synergistic epitope engagement [114]. Homomultivalent and biparatopic nanobodies exemplify distinct yet complementary approaches to combat SARS-CoV-2. Homomultivalent constructs, such as EE (dimeric) and EEE (trimeric), targeting the ACE2-binding interface (interface E), achieve marked improvements in affinity (KD: 930 pM and 520 pM, respectively) and neutralization potency (IC_50_ reductions of 65- to 350-fold compared to monomeric VHH E). While these designs excel in blocking receptor engagement through avidity effects, their reliance on a single epitope renders them susceptible to viral escape mutations like S494P. In contrast, biparatopic nanobodies (e.g., VE, EV) combine non-overlapping epitopes (interface E and UVW), synergistically enhancing neutralization (15–20-fold IC_50_ improvements) while leveraging dual mechanisms, ACE2 competition and premature spike activation, to irreversibly inactivate virions. This dual targeting minimizes escape risks, as mutations in one epitope are counterbalanced by residual binding to the other. However, biparatopic designs demand precise structural engineering to optimize linker flexibility and spatial compatibility.

Recent advances highlight the potential of bispecific nanobodies to overcome viral evolution. A landmark example is the bn03 bispecific nanobody developed by Li et al., which targets two conserved epitopes on the RBD trimer [72]. One arm binds a cryptic epitope (n3130v) on the inner surface of the RBD trimer, while the other targets an outer epitope (n3113v) (Figure 2B). Cryo-EM studies revealed that n3130v binding pulls the RBD closer to the trimer axis, exposing n3113v’s epitope and enabling simultaneous engagement. This dual-binding mechanism achieves broad neutralization against Omicron (including BA.1, BA.2, and BA.4/5) and other VOCs by circumventing mutations in the receptor-binding motif (RBM), which typically evade conventional monoclonal antibodies. The bn03 bispecific design also enhances stability and neutralization potency, as both epitopes exhibit high evolutionary conservation. Beyond neutralization breadth, nanobodies’ small size and stability enable innovative delivery platforms. The bn03 construct has been adapted into inhalable formulations for direct lung delivery, maximizing local antiviral activity while minimizing systemic exposure. Preclinical studies in murine models demonstrated that aerosolized bn03 reduces viral loads in both upper and lower respiratory tracts, even in severe infection scenarios. This approach aligns with broader efforts to repurpose multiepitope nanobodies as inhaled therapeutics, leveraging their rapid tissue penetration and resistance to mucosal degradation.

### 8.2. Nanobodies Binding to Different Epitopes of the Influenza Virus Hemagglutinin (HA) Protein

Influenza A (IAV) and B viruses cause seasonal epidemics, with high-risk strains like H5N1 and H7N9 posing persistent threats due to antigenic drift in the HA protein. Conventional vaccines struggle to address rapid viral evolution, especially in vulnerable populations. While broadly neutralizing antibodies (bnAbs) targeting conserved HA epitopes show promise, limitations such as narrow cross-reactivity and short-lived efficacy hinder their clinical utility.

To overcome these challenges, Laursen et al. engineered multidomain antibodies (MDAbs)—fusion constructs of camelid nanobodies targeting multiple conserved HA epitopes [115]. For instance, MD3606 combines four nanobodies binding both influenza A/B HA stems (SD38/SD36 for influenza A groups 1/2, SD83 for influenza B) and the receptor-binding site (RBS) of influenza B (SD84) (Figure 5A), while MD2407 links these four nanobodies to target both stem and RBS epitopes. Delivered intranasally via AAV vectors in mouse models, these MDAbs achieved long-term protection against H7N9 and H1N1 by simultaneously blocking viral attachment (via RBS binding) and fusion (through stem stabilization). Their modular design enables scalable production and adaptability to emerging strains.

Immunodominant HA head epitopes often overshadow conserved stem-targeting responses. Chen et al. addressed this by isolating E10, a nanobody binding a cryptic, conserved “lateral patch” on the HA head [116]. Structural analysis revealed E10’s epitope centers around residues K166 and S167 (H3 numbering) within the HA1 domain, a region conserved across H1, H3, and H7 subtypes.

These advances highlight nanobodies’ unique advantages in influenza virus research: epitope accessibility (targeting sterically hindered regions), multivalency engineering, and mucosal delivery compatibility. By integrating stem- and head-targeting strategies, next-generation nanobody cocktails could overcome immunodominance biases and provide durable, pan-influenza protection.

### 8.3. Nanobodies Targeting Multiple Epitopes on GFP (Green Fluorescent Protein) and RFP (Red Fluorescent Protein) mCherry

Fluorescent proteins, such as GFP and RFP, have long been central to early nanobody research due to their advantageous properties for scientific investigation. Their inherent stability, straightforward detectability, and utility as markers in live-cell imaging make them ideal targets for developing nanobody-based detection and manipulation tools. Building on this foundation, researchers have engineered nanobodies targeting diverse epitopes across GFP and RFP to enhance experimental versatility.

Structural studies now reveal striking insights into this work: dozens of PDB entries document GFP and its derivative protein–nanobody complex structures. Despite the sequence diversity of these nanobodies, their binding sites frequently overlap. Through systematic analysis of reported GFP-binding nanobodies—including those deposited in the PDB without associated publications (hereafter cited by PDB ID)—we systematically summarized 10 distinct GFP epitopes targeted by these nanobodies (Figure 5B).

Epitope 1 (shown in red, PDB: 3K1K, 3OGO, 6XZF): This is represented by the GFP-enhancer nanobody [54,117].

Epitope 2 (shown in orange, PDB: 3G9A): This is represented by the GFP-minimizer nanobody [54].

Epitope 3 (shown in pink, PDB: 6LR7, 7E53, 7SAH, 7SAI, 8SFS, 8SLC): This is represented by the LaG16 and LaG30 nanobody [118,119,120], and it is the most frequently targeted GFP epitope in PDB entries; a lot of nanobodies bind this region.

Epitope 4 (shown in cyan, PDB: 7CZ0): This is the binding site for the Sb92 nanobody. While it is structurally close to Epitope 3, the nanobody’s orientation differs significantly.

Epitope 5 (shown in yellow, PDB: 8XLD): This is the only nanobody documented to bind the top of the GFP barrel structure; all others bind the side walls.

Epitope 6 (shown in limon, PDB: 6LZ2): This is a unique binding site distinct from others [121].

Epitope 7 (shown in deep teal, PDB: 8G0I): This closely resembles the binding site of Epitope 6 (6LZ2) but with distinct interactions [122].

Epitope 8 (shown in marine, PDB: 8SG3, 8SFV, 8SFX): This is shared by three nanobodies with similar binding modes.

Epitope 9 (shown in purple, PDB: 8HGI): This is the sole shark-derived V_NAR_ targeting GFP reported to date [123].

Epitope 10 (shown in blue, PDB: 8SFZ): This is proximal to the GFP-minimizer epitope but with a completely different binding mode.

Next, we highlight applications of selected GFP-targeting nanobodies, which have enabled the development of diverse tools, including biosensors, affinity reagents for protein purification, and imaging and cellular manipulation systems. Nanobodies binding to GFP have been engineered to precisely modulate its fluorescence properties, either enhancing or suppressing brightness through epitope-specific interactions. These nanobodies induce conformational changes in GFP, altering its chromophore microenvironment and thereby modifying absorption spectra. For instance, the GFP-enhancer (GBP1) increases fluorescence intensity by up to four-fold (PDB: 3K1K), while the GFP-minimizer (GBP4) reduces fluorescence by five-fold (PDB: 3G9A) [54]. Structural analysis revealed that these nanobodies induce subtle changes in the chromophore environment, affecting the protein’s absorption properties. Additionally, nanobody-induced fluorescence modulation has been applied in vivo, where it has been used to monitor protein expression, subcellular localization, and dynamic processes like the tamoxifen-induced translocation of the estrogen receptor [54]. This demonstrates the potential of nanobodies to manipulate protein conformations and provide new tools for biological research. Another study demonstrated that the crystal structure of GFPuv in complex with the anti-GFP nanobody LaG16 revealed its ability to simultaneously bind GFP alongside another nanobody, GFP-enhancer, at distinct sites [118]. Leveraging this structural insight, researchers engineered a chimeric nanobody with an optimized linker, resulting in ultra-high affinity and enhanced efficiency in purifying GFP-tagged proteins [119].

RFP mCherry-specific nanobodies, particularly those from the LaM family (LaM1, LaM2, LaM3, LaM4, LaM6, and LaM8) developed by Fridy et al. [124], have become some of the most widely used tools in nanobody research due to their well-characterized structures [119,125,126]. Figure 5C illustrates the relative binding positions of these nanobodies to mCherry, revealing that despite their sequence diversity, LaM2, LaM3, and LaM6 target closely overlapping epitopes on mCherry. These nanobodies, based on their structural properties, have been engineered to develop a range of novel applications. One such application involves the fusion of nanobodies with plant immune receptors to enhance disease resistance. Researchers utilized the versatility of mammalian antibodies to modify plant nucleotide-binding, leucine-rich repeat immune receptors (NLRs) [127]. By integrating camelid nanobodies that recognize fluorescent proteins, these chimeric receptors enabled plants to mount immune responses upon the introduction of fluorescent proteins. This approach provides a pathway to rapidly develop pathogen-resistant crops, offering a promising strategy for enhancing food security.

Another significant application of RFP nanobodies is in optogenetic control, where nanobodies are engineered to regulate their binding to targets through light activation [128]. Gil et al. developed a strategy by inserting the AsLOV2 domain into the flexible loop of nanobodies, which enabled light-induced modulation of their binding affinity [129]. This technique was successfully applied to control nanobodies targeting mCherry, GFP, and actin. Interestingly, two distinct modes of regulation were observed, where light exposure either activated or deactivated the nanobody binding. The structural analysis of the AK74 construct revealed that the AsLOV2 domain could act as a light-switchable mechanism, where the transition from dark to illuminated states induced conformational changes, either preventing or allowing the nanobody to bind to its target. This optogenetic strategy opens new avenues for dynamically controlling protein localization and interactions in living cells, though further optimization is needed to enhance its dynamic range and affinity differences.

## 9. Leveraging Artificial Intelligence (AI) in VHH Engineering

Nanobodies exhibit exceptional versatility by binding to multiple epitopes on the same antigen, a strategy that not only enhances neutralization and specificity but also allows these nanobodies to be used as controllable modules for precise antigen manipulation. This multiepitope targeting approach has shown significant promise in various applications, from neutralizing complex viral particles like SARS-CoV-2 to modulating the activity of fluorescent proteins. However, the design and optimization of such nanobodies face challenges in terms of affinity, specificity, and in vivo stability, which are critical for their therapeutic and diagnostic efficacy.

To address these challenges, researchers are increasingly adopting computational methods, especially AI-driven approaches, which provide new possibilities for nanobody engineering. AI tools can quickly predict nanobody structures, enhance binding affinities, and design new nanobodies with specific functions, thus speeding up development and lessening the reliance on time-consuming trial-and-error experiments [41,130,131,132,133,134,135]. Recent advances in machine learning and deep learning, particularly in VHH structure determination and design, coupled with the 2024 Nobel Prize in Chemistry awarded to David Baker, Demis Hassabis, and John Jumper for their pioneering work in AI-driven protein structure prediction and design, signal significant potential for computational solutions to these challenges. This section highlights recent AI-based advances in nanobody structure prediction, humanization, and design [136,137].

### 9.1. In Silico Nanobody Structure Determination

Since the number of applications utilizing nanobodies is increasing continuously, the demands for the protein engineering of VHHs to enhance their physical and chemical properties are never satisfied. However, due to reliable three-dimensional models of nanobodies being critical for almost all the applications related to nanobody engineering, a cost-effective way for determining nanobody structures and modeling the CDRs, especially the CDR3 region, is required. To overcome the limitations in determining nanobody structures, several AI tools have recently been developed.

### 9.2. AI Tools for Computing Individual VHH Structures

Prior methods for determining protein tertiary structure have employed comparative modeling, utilizing a related structure as a guide, and then optimizing the resulting model through energy minimization [138,139,140,141,142,143]. AlphaFold2 (AF2), DeepMind’s groundbreaking artificial intelligence program, utilizes multiple sequence alignments (MSAs) and attention mechanisms to achieve high-accuracy protein structure prediction, which can be applied to model nanobody structures accurately [23]. This pivotal invention marks a starting point for the creation of sophisticated computational tools for determining the three-dimensional conformations of antibodies and VHHs.

DeepAb, a deep learning method developed for accurate antibody Fv structure prediction from sequence, can also be applied to nanobody modeling. However, while nanobody CDRs share similarities with antibody Fv CDR H3 loops, further model refinement is necessary for optimal nanobody structure prediction. Furthermore, DeepAb exhibits relatively slow processing speeds (approximately 10 min per sequence), does not effectively incorporate template information, and provides limited quality assessment metrics, which limit its use in accurate nanobody structure building [144].

ABlooper is an end-to-end equivariant deep learning-based CDR loop structure prediction tool, which produces antibody models of similar accuracy to both AF2 and DeepAb on a far faster timescale. The drawbacks are that it relies on external tools for framework modeling and it does not support nanobody modeling [145].

NanoNet is an accurate end-to-end deep learning-based method supporting high-throughput 3D nanobody modeling. Considering the fact that previously existing tools mainly took mAb or T cell receptor (TCR) modeling into consideration [146,147,148], NanoNet is the first tool tailored for nanobody structure determination. It achieves balance in speed and accuracy when compared with AF2 or DeepAb [149].

Since prediction methods based on MSA, like AF2, require a time-consuming process to construct a meaningful MSA and spend long runtimes predicting protein structure, predicting protein structures under high throughput is computationally prohibitive for many users [150,151,152,153]. IgFold, without applying MSA, however, is a fast and accurate model that specializes in the prediction of antibody structures comparable to the accuracy of AF2 with significantly less time. It leverages embeddings from AntiBERTy [154] to directly predict the atomic coordinates that define the antibody structure, which robustly incorporates template structures and supports nanobody modeling [155].

ImmuneBuilder is a set of deep learning models developed to predict the structure of proteins of the immune system. It contains three models including ABodyBuilder2, NbBuilder2 and TCRBuilder2. Among them, NbBuilder2 is a nanobody-specific model which generates structures of nanobodies with an accuracy comparable to that of AF2, but with a prediction speed that is a hundred times faster [156].

### 9.3. Predicting VHH Paratopes Utilizing AI

While modeling individual nanobody structures is essential for design and engineering, understanding their interactions with target antigens is crucial for predicting efficacy and developing therapeutic applications. With larger antibody sequence datasets now available, paratope prediction has become more effective.

Parapred is the first application of deep learning techniques to the paratope prediction problem by using only antibody sequence stretches corresponding to the CDRs with two extra residues on the either side as input. The model does not rely on any higher-level antibody features like full sequence, homology model, crystal structure or antigen information [157].

Compared to the CNN- and RNN-based Parapred, ParaAntiProt, a recently proposed model incorporating various pre-trained antibody and protein language models, exhibits enhanced performance in paratope prediction by leveraging transfer learning for feature extraction alongside a CNN block. Besides delivering accurate antibody paratope predictions, the model demonstrates strong performance in predicting nanobody paratopes [158].

### 9.4. Computational Tools for Nanobody Docking and Screening

Besides paratope prediction, the computational prediction of how antigens and nanobodies dock is a critical area of interest, and artificial intelligence has yielded notable breakthroughs in this regard [159,160,161,162,163]. NbX is the first machine learning-guided re-ranking method for the accurate prediction of native-like nanobody poses since previous methods were developed on a majority of conventional Ab-Ags and general protein–protein interactions, suggesting potential biases [164].

DLAB is a structure-based deep learning approach for the early-stage virtual screening of antibody therapeutics, especially when an epitope of an antigen is known but no suitable antibodies targeting that epitope have been found [165].

### 9.5. State-of-the-Art Structural Modeling Utilizing AlphaFold3

Building upon the success of AlphaFold2, AlphaFold3 (AF3) introduces a transformative diffusion-based architecture that dramatically expands the scope of structural prediction. Unlike its predecessor, AF3 can predict the joint structures of complex assemblies involving proteins, nucleic acids, small molecules, ions, and modified residues. This capability, coupled with a streamlined process that reduces MSA processing and directly predicts atomic coordinates via a diffusion module, significantly enhances the modeling of challenging systems such as antibody–antigen and nanobody–antigen complexes. The use of multiple model seeds further refines these predictions, establishing AF3 as a major advancement in the field of structural biology with far-reaching implications [5].

### 9.6. Potential Limitations in Computational Structure Determination

Notwithstanding the impressive advances AI has enabled in nanobody structural modeling, there are still areas requiring substantial improvement. A recent study has evaluated six state-of-the-art AI programs in modeling nanobody structures [166], including AF2 [23], ESMFold [167], IgFold [155], Nanonet [149], Yang-server [168] and OmegaFold [169]. This confirms that, though the modeling of framework regions of nanobodies is consistently good in all cases, CDR modeling, especially for CDR3, still remains a great challenge. Among all the tested models, only OmegaFold satisfied the 2.5 Å RMSD threshold for over 50% of the picked CDR3 structures. Surprisingly, NanoNet, which was designed for high-throughput 3D nanobody modeling, only achieved a 25% accuracy. The results suggest that though there have been substantial advances, the accuracy of the generated models is still limited and nanobody modeling remains a challenge.

Two recent studies have pointed to specific limitations in AF3’s predictive capabilities [137,170]. One has tested the capability of AF3 to capture the fine details and interplay between antibody structure prediction and antigen docking accuracy. AF3 achieves a high-accuracy docking success rate compared to that of previous methods for antibodies and nanobodies, which lies from 0 to 8.9% and 13.4%, respectively. Furthermore, the accuracy of the model was reported to increase to 60% when evaluating 1000 seeds in non-reproducible work. Most impressively, AF3 docking success is independent of loop length for both antibodies and nanobodies. However, when applying only a single seed, there remains room for improvement of the 60% failure rate for both antibody and nanobody docking for AF3 [170]. Another recent study critically evaluates the nanobody epitope prediction performance of AF3, offering insights into its strengths and limitations. By using a representative and diverse benchmarking dataset of 70 nanobody–antigen complexes, AF3 was shown to successfully identify the binding region, with an overall success rate still falling below 50%. Typically, with longer CDR3 loop length, flexibility makes predicting a single bound conformation more challenging. It has also been found that beyond the length of CDR3, its folding orientation, whether stretched or kinked, significantly influences the success of epitope prediction, with a success rate exceeding 88% with a stretched CDR3 conformation and a success rate of only nearly 20% for a kinked CDR3 structure [137]. These two studies suggest that, despite AF3’s advancements in modeling complex molecular interactions, limitations persist.

While significant progress has been made in computational nanobody modeling, driven by advancements in AI and deep learning, several key challenges remain. Methods like AF2, DeepAb, ABlooper, NanoNet, IgFold, and ImmuneBuilder, each with their own strengths and limitations, have significantly improved nanobody structure prediction, particularly for framework regions. However, accurate modeling of CDRs, especially the highly variable CDR3 loop, continues to be a major hurdle. Even the powerful AF3, with its expanded capabilities for modeling complex interactions, faces limitations in accurately predicting nanobody–antigen docking and epitope prediction, particularly for non-canonical CDR3 conformations. However, with more and more antibody and VHH structures becoming experimentally available, further research and development can be carried out to enhance the accuracy and reliability of these computational tools, especially for challenging aspects like CDR modeling and interaction prediction, to fully realize the potential of nanobodies in therapeutic and biotechnological applications. A brief summary of all the above models for computing structures can be seen in Table 4.

## 10. Computational Humanization Strategies Tailored for Nanobodies

The development of an immune response against a therapeutic antibody, specifically the generation of anti-drug antibodies (ADAs), can diminish treatment effectiveness and pose risks to patient health [173,174,175,176,177,178]. Therefore, as mentioned in the above sections, mitigating this potential hazard is crucial prior to initiating clinical trials. Given this context, it is logical to hypothesize that reducing the immunogenicity of a non-human derived antibody could be achieved by increasing its resemblance to naturally occurring human antibodies, which are well tolerated by the human immune system. This concept has driven the development of various antibody humanization technologies. While traditional antibody and nanobody humanization relies on experimental methods, which are heavily dependent on the specific clone and involve extensive, time-consuming, and costly trial-and-error through amino acid mutations, integrating computational approaches for a more rational and efficient process is warranted [179].

Because monoclonal antibodies (mAbs) are more established biotherapeutics than nanobodies and thus have larger, more readily accessible benchmark datasets, the development of in silico methods for humanization has primarily focused on conventional antibodies [180,181,182,183,184]. For example, Hu-mAb—a web-based computational tool that uses a random forest (RF) classifier to systematically humanize VH and VL sequences by suggesting humanness-enhancing mutations—is designed exclusively for murine precursor sequences. Consequently, it does not support the humanization of alternative antibody formats such as nanobodies and asymmetric antibodies [185]. This phenomenon is attributable to the hydrophilicity-enhancing FERG residues in the FR2 regions, which are distinctive for camelid nanobodies, as we have mentioned above (Figure 3B,C) [4,186,187,188,189]. These residues contribute to their increased solubility due to enhanced hydrophilicity [13]. Furthermore, nanobodies more frequently incorporate framework residues into their paratopes, rendering previous computational methods—which assume that only CDRs contribute to antigen binding—inapplicable [190]. As a result, computational approaches that incorporate the characteristic topology of nanobodies are necessary.

Based on big data analysis, Llamanade is the first open-source software designed to facilitate automated, structure-guided, and robust nanobody humanization [191]. It comprises five main modules that carry out nanobody structural modeling, sequence annotation, sequence analysis, and structural analysis, and that generate a humanization score for each humanized nanobody. Input can include not only nanobody sequences but also high-resolution structures of nanobodies or antigen–nanobody complexes. Nanobody sequences are first annotated using the Martin scheme to define FRs and CDRs. Then, these annotated input nanobodies are compared to a position probability matrix generated based on the T20_FR_ score using VH_human_ sequences from the authors’ human IgG library [192]. Candidate residues for mutation are selected if their occurrence is sufficiently low (<10%) at the corresponding position in VH_human_. Subsequently, only solvent-exposed residues that do not participate in backbone intramolecular interactions and do not impact antigen–nanobody interactions are humanized, according to the structural analysis performed by the software. Finally, a humanization score is calculated to quantify the humanness level of the nanobodies. While seemingly comprehensive, Llamanade does not provide a quantitative measurement of the uncertainty of nanobody humanization, which is an area for improvement.

Alongside statistical computational methods, deep learning techniques, though not yet widely adopted, are being utilized in nanobody humanization. AbNatiV is a novel antibody nativeness assessment method based on a Vector Quantized Variational Autoencoder (VQ-VAE) [193]. It evaluates the likelihood of input sequences originating from immune-system-derived antibodies, including human VH and VL domains and camelid VHHs. AbNatiV provides both an interpretable overall score for the complete sequence and a residue-level nativeness profile, which can guide antibody engineering and humanization. Compared to other methods designed for antibody humanization [194,195], AbNatiV demonstrates superior classification performance in distinguishing human antibodies or camelid nanobodies from antibodies of other species. Applying AbNatiV’s humanness and VHH-nativeness analysis to two experimentally screened nanobodies demonstrated that its enhanced sampling pipeline helps retain binding activity and biophysical stability. In contrast, conventional structural and residue frequency-based humanization disrupted both properties in the same nanobodies. Therefore, AbNatiV represents a significant advancement in our ability to humanize nanobodies while preserving in vivo properties that are highly competitive with those of naturally occurring antibodies.

The most recent work presented by Ma et al. [196] even takes a step forward, introducing an efficient and adaptive diffusion approach called HuDiff, specifically designed for the effective humanization of antibodies and nanobodies. The model has two sub-models: HuDiff-Ab for antibodies and HuDiff-Nb for nanobodies. Both share a similar architecture, consisting of encoders, hidden blocks, and a decoder. Interestingly, the model requires only the input of CDR sequences and works exclusively at the sequence level during humanization. This makes it more applicable than previous methods, such as Llamanade [191] and AbNatiV [193], which rely on the whole sequence of an original nanobody and incorporate a structure prediction stage in their workflow. To validate HuDiff-Nb, the authors selected 3-2A2-4 as a template, which targets the SARS-CoV-2 RBD antigen. Surprisingly, the top humanized nanobody among the five selected top hits even demonstrated improved binding affinity compared to the parental alpaca nanobody 3-2A2-4, exhibiting a K_D_ value of 2.52 nM compared to 5.47 nM, respectively, in a biolayer interferometry (BLI) experiment. Based on this exciting result, HuDiff may serve as a valuable tool for researchers advancing antibody and nanobody humanization.

Despite the apparent breakthroughs described above, it should be noted that the scarcity of publicly available nanobody humanization models and the relatively limited and dispersed nature of nanobody data [197,198,199,200] still hinder the development of new computational methods. More generally, the field currently relies on a specific definition of “humanness” upon which all present nanobody humanization models are based. This definition is challenged by the fact that even genetically human antibodies can still provoke anti-drug antibody (ADA) production [173]. Future work should take all of these factors into consideration to achieve greater progress.

## 11. De Novo AI-Based Nanobody Design

De novo protein design, defined as navigating the sequence fitness landscape to identify proteins with desired properties and divergent sequences compared to naturally occurring counterparts, has been a long-standing challenge and a focus of sustained research within the scientific community [201,202,203,204,205,206]. For over two decades, tools such as Rosetta have successfully facilitated in silico protein engineering using traditional de novo design methods [207]. These methods rely on physics-based modeling and iterative searches to explore protein sequence space by simulating the biophysical interactions that determine sequence properties [208]. While a deeper understanding of protein physics could theoretically inform the design of novel proteins tailored to specific functions, the complexity of these interactions and the computational cost of exhaustive physics-based modeling inevitably limit the exploration of diverse functional protein variants. Consequently, computational models leveraging generative artificial intelligence have emerged as a promising alternative in the past decade. A comprehensive overview of general de novo protein design using generative models is provided in the recent work of Winnifrith et al. [205]. Here, we pay our attention to the recent progress in de novo nanobody design.

A 2021 publication introduced the first alignment-free model for state-of-the-art mutation effect prediction without reliance on experimental data, applying it at scale to nanobody sequence design. By training an autoregressive generative model on naive nanobody repertoires, Shin et al. designed and experimentally validated a diverse library of 10^5^ nanobodies. This library demonstrated nearly 2-fold higher expression levels compared to a 1000-fold larger synthetic library. Moreover, the designed sequences exhibited greater divergence than those observed in natural repertoires [209]. While not focused on designing specific nanobodies targeting defined epitopes, this approach, by generating libraries with increased diversity, facilitates rapid, efficient, and cost-effective discovery of nanobody candidates, minimizing library waste and experimental effort, and providing an ideal starting point for affinity maturation to enhance binding.

Regarding generative models specifically tailored for de novo nanobody design, the first, to our knowledge, was developed by David Baker’s group (Figure 6). Building upon their previous diffusion model for general de novo protein design, RFdiffusion [210], Baker et al. developed a fine-tuned network specifically for designing de novo nanobodies targeting user-specified epitopes [211]. This refined model, trained on antibody complex structures, allows for the specification of framework structure and sequence during inference. By providing the framework structure in a global-frame-invariant manner during training, the model can identify target protein residues interacting with CDR loops, enabling the design of novel CDR loops that target the specified epitope. Subsequently, ProteinMPNN [6], another model developed by Baker et al. for sequence design from protein structures, is used to generate the CDR loop sequences. Following this, a fine-tuned model, RoseTTAFold2 [212], is employed to eliminate decoy VHHs. Experimentally, using a fixed humanized VHH framework mentioned above [4], nanobodies were designed targeting TcdB, Influenza HA, RSV site III, SARS-CoV-2 RBD, and IL-7Rα, demonstrating moderate affinity and high specificity.

Another recently submitted study proposes IgGM, a generative model combining a diffusion model and a consistency model for generating antibodies and nanobodies with functional specificity. This model employs a multi-level network architecture comprising a pre-trained language model for sequence feature extraction, a feature learning module for identifying pertinent features, and a prediction module that generates designed antibody and nanobody sequences along with predicted antibody–antigen and nanobody–antigen complex structures. Computational experiments demonstrate that IgGM achieves superior performance across multiple design tasks. Nevertheless, further experimental validation carried out in wet lab is required to confirm its efficacy in designing high-affinity nanobodies [213].

As demonstrated, few methods are currently tailored for de novo nanobody design. We therefore anticipate that future models will leverage emerging AI techniques and the growing body of nanobody structural and sequence data to design improved, more specific, and even multifunctional nanobodies for diverse applications.

## 12. Perspective

Looking forward, the use of AI and big data will continue to drive the evolution of nanobodies. As more structural and sequence data become available, AI models will be able to predict and design multifunctional nanobodies that can target a wide range of diseases, from cancer to autoimmune disorders. The combination of computational power and experimental validation is expected to accelerate the development of next-generation nanobodies with greater therapeutic potential. Furthermore, new advancements in nanobody humanization, in vivo stability, and engineered nanobody libraries will continue to push the boundaries of nanobody applications in medicine and biotechnology.

## 13. Conclusions

Nanobodies have evolved from basic research to clinically validated therapeutics, driven by structural insights, advanced engineering, and AI-driven innovation. Their versatility in targeting traditionally inaccessible epitopes, combined with AI’s ability to optimize design and reduce development timelines, positions them as powerful tools for addressing unmet biomedical needs. Future advances will likely focus on improving humanization efficiency, enhancing multivalency and stability, and expanding applications in emerging areas like pandemic response and precision medicine. As computational methods and structural data grow, nanobodies are poised to become even more integral to next-generation diagnostics, therapies, and biotechnological solutions.

## Figures and Tables

**Figure 1 biology-14-00547-f001:**
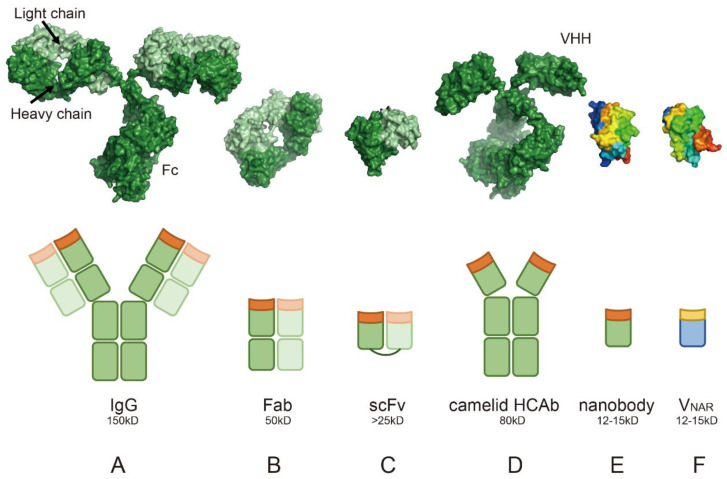
Structural and size comparison of different antibodies The upper panel depicts antibody structures or fragments based on representative crystal structures, while the lower panel provides simplified schematic diagrams for clarity, with orange, light orange and yellow indicating the CDR of the respective antibody form. From left to right: (**A**) IgG (traditional antibody): full-length immunoglobulin G (PDB: 1IGT); green indicates heavy chains, light green indicates light chains. (**B**) Fab fragment: antigen-binding fragment (PDB: 1MLC); green indicates the Fab region of the heavy chain, light green indicates the light chain. (**C**) scFv (single-chain variable fragment): linked variable heavy and light chains (PDB: 1DZB); green highlights the variable domain of the heavy chain, and light green highlights the variable domain of the light chain. (**D**) Camelid heavy-chain only antibody (HCAb): single-chain antibody derived from camelids, with the VHH region (PDB:1ZVH) aligned and grafted onto the IgG framework (PDB:1IGT); the two upper arms splayed out indicate the VHH domain. (**E**) Nanobody (VHH): isolated VHH domain (PDB: 3K1K) in the upper panel, rainbow-colored, with N-terminus blue to C-terminus red. (**F**) Shark-derived V_NAR_: single-domain antibody from nurse shark (PDB: 1T6V) in the upper panel, rainbow-colored, with N-terminus blue to C-terminus red.

**Figure 2 biology-14-00547-f002:**
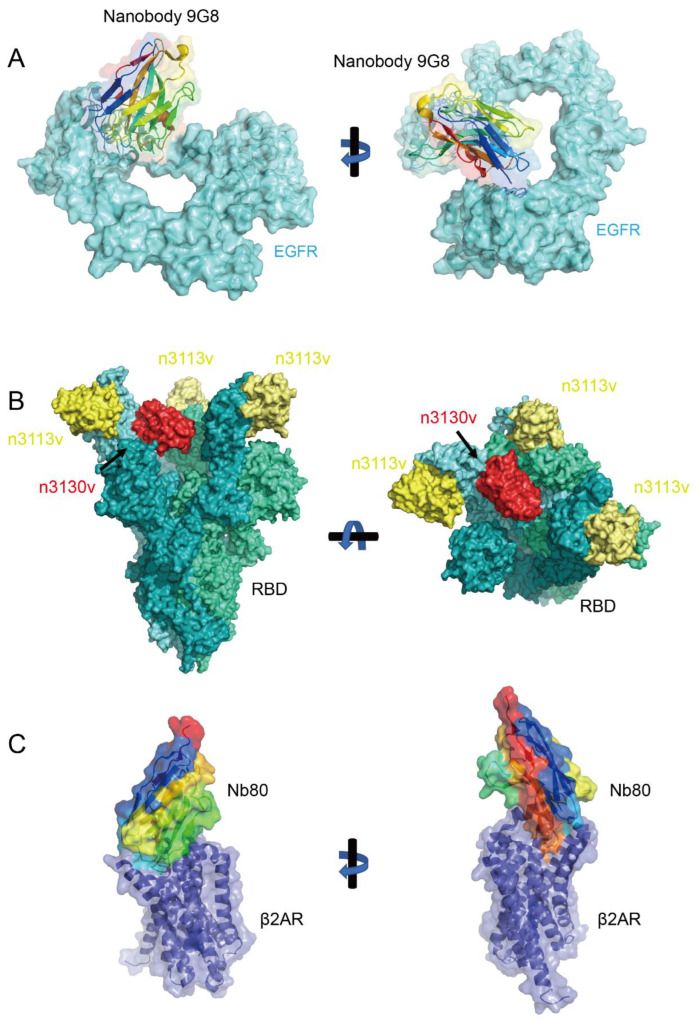
Nanobody advantages in antigen binding (**A**) Concave surface binding (PDB:4KRP). The nanobody 9G8 (rainbow-colored, N-terminus blue to C-terminus red) binds the concave extracellular region of the epidermal growth factor receptor (EGFR, shown in cyan). Its compact size enables access to deep hydrophobic pockets inaccessible to conventional antibodies due to size constraints. (**B**) Cryptic epitope recognition (PDB:7WHI). Three nanobodies n3113v (yellow) bind the outer surface of the receptor-binding domain (RBD, three conformations in green), while nanobody n3130v (red) targets a cryptic epitope on the RBD’s interior. This hidden binding site becomes exposed under specific conditions, demonstrating nanobodies’ ability to overcome the size/rigidity limitations of traditional antibodies. (**C**) Dynamic region stabilization (PDB:3P0G). Nanobody Nb80 (rainbow-colored, N-terminus blue to C-terminus red) stabilizes the β2-adrenergic receptor (β2AR, shown in blue) in its active state.

**Figure 3 biology-14-00547-f003:**
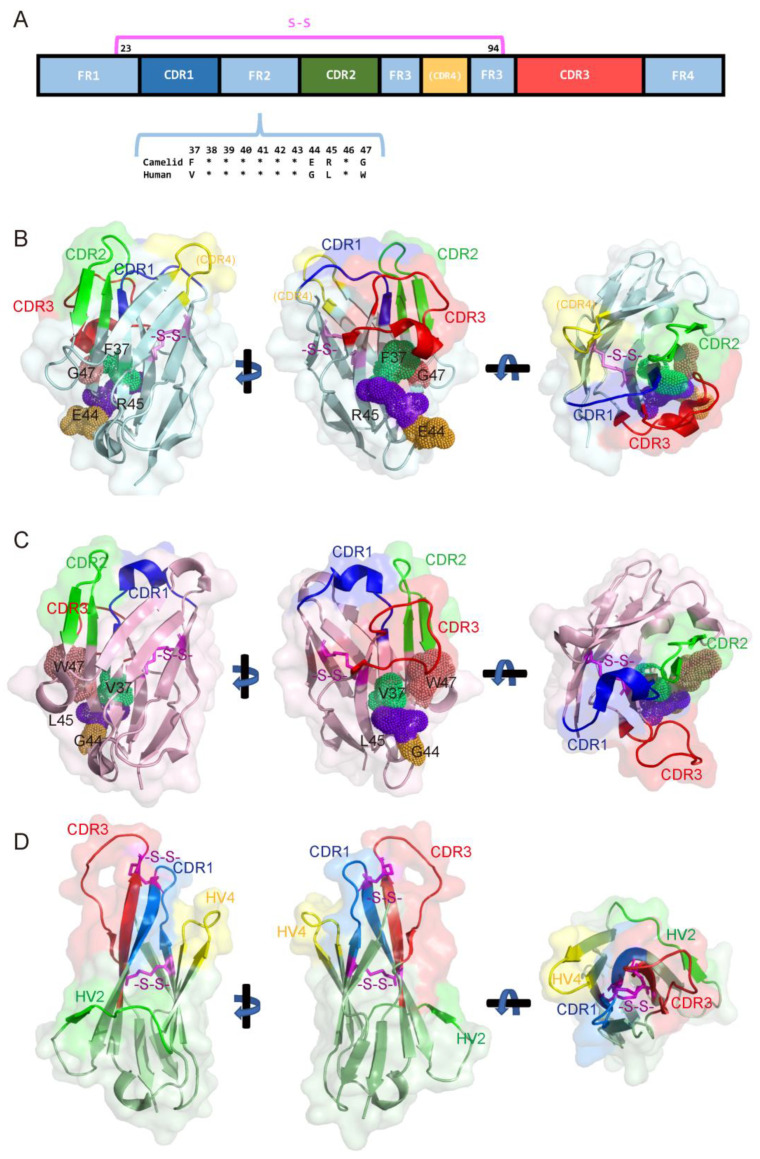
Comparison of camelid-derived (VHH), human-derived (VH), and shark-derived (V_NAR_) nanobodies (**A**) Schematic of domain architecture for camelid- and human-derived nanobodies. Framework regions (FR1-FR4) are shown in light blue, with blue for CDR1, green for CDR2, red for CDR3, and yellow for a putative CDR4 (embedded within FR3). Pink indicates intra-nanobody disulfide bond positions. Camelid and human-derived nanobodies exhibit significant differences in four conserved amino acids (positions 37, 44, 45, and 47) within the FR2 region. The “*” represents any amino acid, while the uppercase letter is the single-letter abbreviation of the amino acid code. (**B**) A 3D structure of a typical camelid-derived nanobody (PDB: 1OP9). The color coding follows panel A; FRs are displayed in pale cyan, with conserved amino acids (F37 in lime green, E44 in bright orange, R45 in purple blue and G47 in salmon) in FR2 highlighted as dot-mode side chains to emphasize their structural roles. CDRs, FRs, and disulfide bonds are colored as described in panel A. (**C**) A 3D structure of a typical human antibody heavy-chain variable domain (VH) (PDB: 4U3X). The color coding follows panel A; FRs are displayed in light pink, with conserved amino acids (V37 in lime green, G44 in bright orange, L45 in purple blue and W47 in salmon) in FR2 highlighted as dot-mode side chains to emphasize their structural roles. CDRs and disulfide bonds are colored as described in panel A. (**D**) A 3D structure of a shark-derived type 2 V_NAR_ (PDB: 2COQ). This structure features an additional disulfide bond linking CDR1 and CDR3, stabilizing the CDR architecture. CDR1 and CDR3, as well as disulfide bonds, follow the color scheme of panel A. The hypervariable regions HV2 and HV4 are shown in green and yellow, respectively, while FRs are displayed in pale green. The extra disulfide bond enhances structural rigidity compared to camelid VHHs.

**Figure 4 biology-14-00547-f004:**
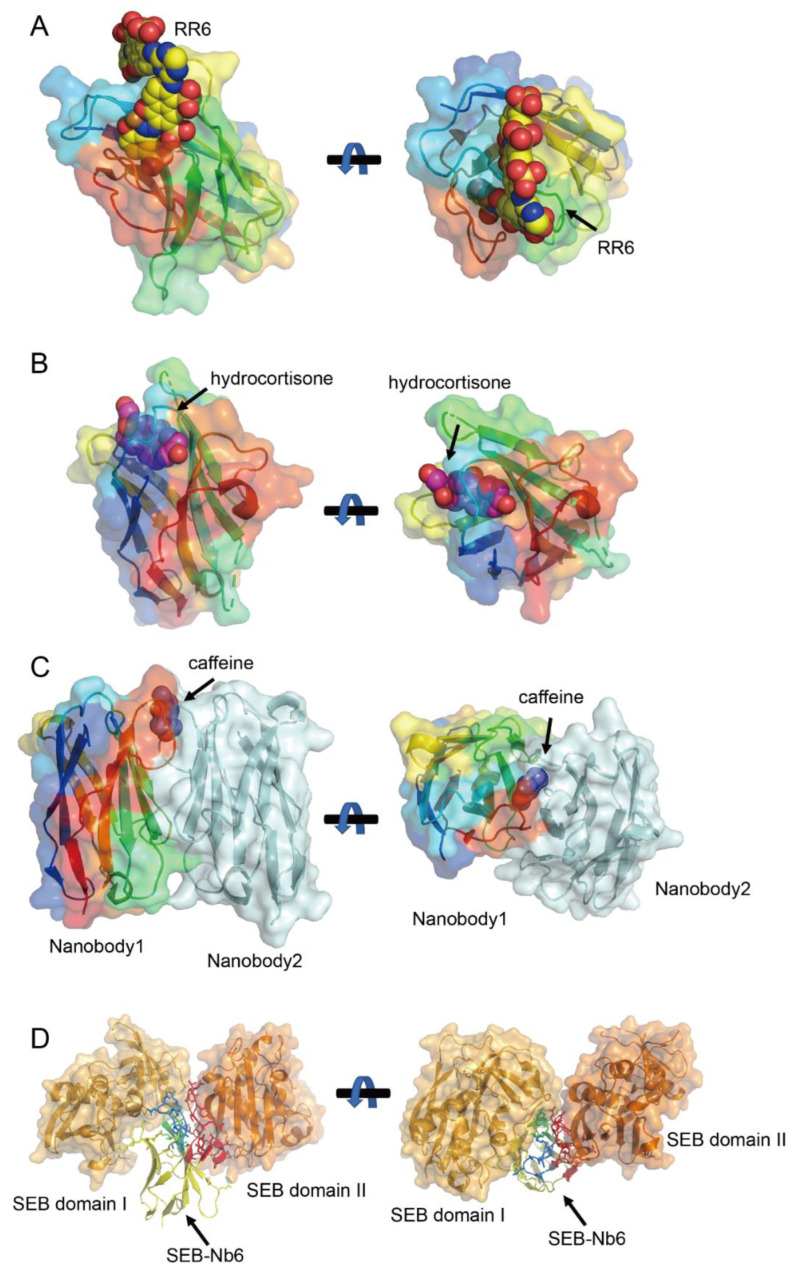
Unique nanobody binding modes (**A**) Traditional multi-CDR binding to small antigen (PDB: 1QD0). The nanobody (rainbow-colored, N-terminus purple to C-terminus red) binds RR6 (yellow) using a conventional multi-CDR interaction, where multiple CDR loops collectively recognize the antigen’s surface. (**B**) Tunnel-mode binding to small antigen (PDB: 6ITQ). The nanobody (rainbow-colored, N-terminus purple to C-terminus red) encapsulates hydrocortisone (pink) via a tunnel-like configuration. CDR1 (blue) forms the primary structural scaffold, tightly enclosing the steroid molecule. (**C**) Dimeric binding to a small antigen (PDB: 6QTL). A nanobody dimer (nanobody1 in rainbow colors, N-terminus purple to C-terminus red; nanobody2 in palecyan) binds caffeine (blue) through a cooperative interface. The dimeric arrangement enhances binding avidity by sandwiching the antigen between the two nanobody molecules. (**D**) Intrinsic dual-epitope engagement (PDB: 8YBM). The nanobody (backbone in yellow, CDR1 in blue, CDR2 in green, CDR3 in red) simultaneously binds two spatially segregated epitopes on Staphylococcal enterotoxin B (SEB), the side chains of residues in non-CDRs forming interactions are also highlighted.

**Figure 5 biology-14-00547-f005:**
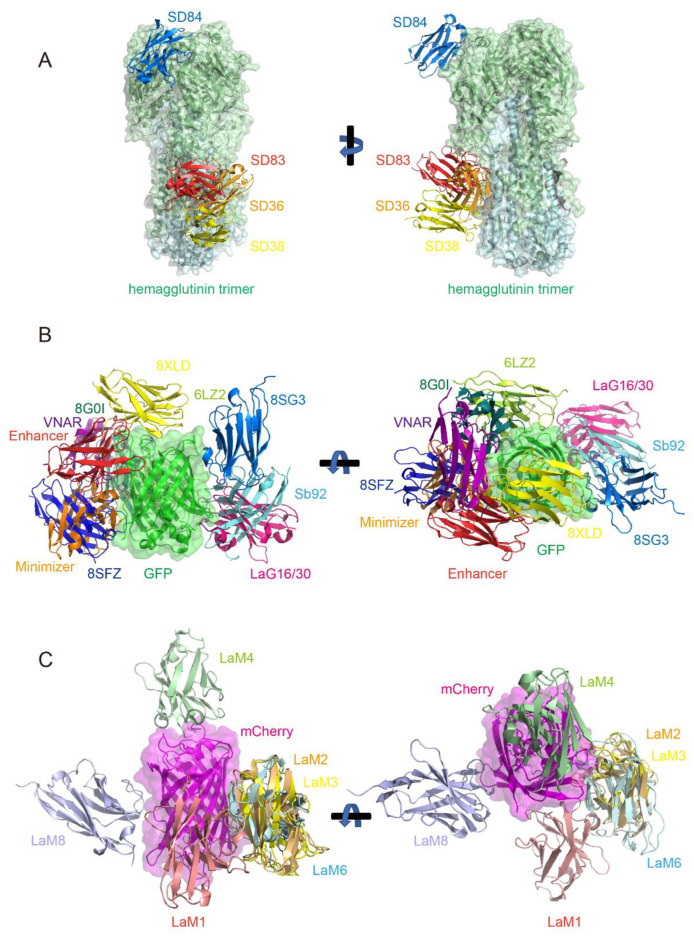
Multiple nanobodies targeting to one antigen (**A**) Multiple nanobodies targeting influenza hemagglutinin (HA) trimer. The HA trimer (green) is shown with four nanobodies binding to distinct epitopes: SD38 (yellow), SD36 (orange), and SD83 (red) cluster near the stem region, recognizing adjacent epitopes; SD84 (blue) binds the receptor-binding site (RBS) at the HA head. (**B**) Nanobody epitope diversity on GFP and mutants. GFP and its variants (green) are depicted with 10 distinct epitopes recognized by nanobodies (colors indicate unique binding sites): Epitope 1 (red, PDB: 3K1K); Epitope 2 (orange, PDB: 3G9A); Epitope 3 (pink, PDB: 6LR7); Epitope 4 (cyan, PDB: 7CZ0); Epitope 5 (yellow, PDB: 8XLD); Epitope 6 (lime green, PDB: 6LZ2); Epitope 7 (teal, PDB: 8G0I); Epitope 8 (marine blue, PDB: 8SG3); Epitope 9 (purple, PDB: 8HGI); Epitope 10 (blue, PDB: 8SFZ). (**C**). Nanobody binding to mCherry’s multiple epitopes. The red fluorescent protein mCherry (magenta) is shown with six nanobodies recognizing distinct sites: LaM1 (salmon, PDB: 8IM1); LaM2 (orange, PDB: 6IR2); LaM3 (yellow, PDB: 8ILX); LaM4 (light green, PDB: 6IR1); LaM6 (sky blue, PDB: 7SAL); LaM8 (lavender, PDB: 8IM0).

**Figure 6 biology-14-00547-f006:**
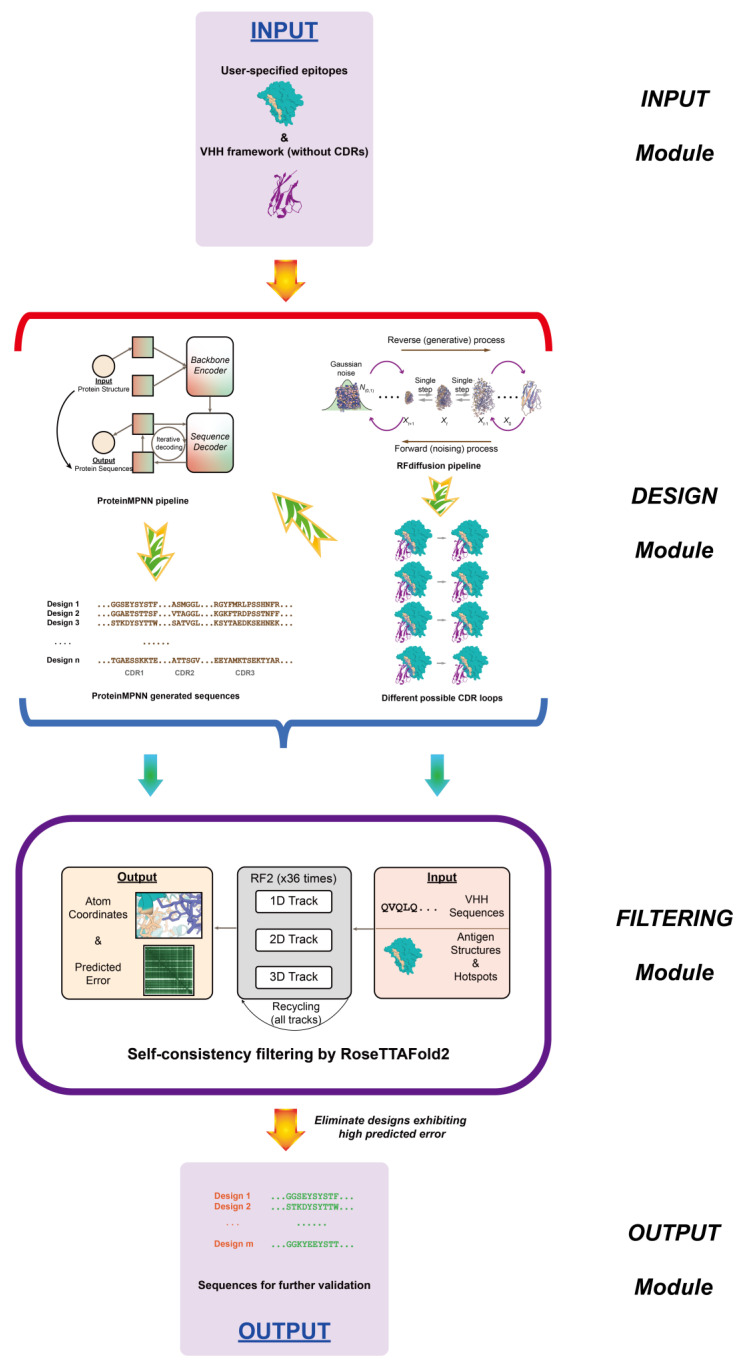
Workflow for computational design of VHHs using a fine-tuned RFdiffusion pipeline The process begins with user-provided target epitopes and a VHH framework lacking complementarity-determining regions (CDRs). RFdiffusion generates an ensemble of potential CDR loop conformations. Subsequently, ProteinMPNN [6] designs corresponding amino acid sequences for each generated structure. To ensure structural integrity and compatibility with the target antigen, a self-consistency filter is applied using RoseTTAFold2 [212]. This filter evaluates predicted alignment error (pAE) and root mean square deviation (RMSD) between the designed VHH and the target VHH–antigen complex. Designs exhibiting low predicted error are selected for downstream experimental validation [211]. All 3D structural models were visualized using Open-Source PyMOL^TM^.

**Table 1 biology-14-00547-t001:** Key development milestones of nanobodies (1993–current).

Year	Milestone	Description	References
1993	Discovery of heavy-chain-only antibodies (HCAbs)	HCAbs were first discovered in camelids (camels, llamas, and alpacas) by C Hamers-Casterman and colleagues, marking the discovery of single-domain antibodies (VHH).	[1]
1996	First nanobody crystal structure	The first crystal structures of nanobodies were solved, providing insights into their binding mechanisms and how the CDR3 loop plays a critical role in their high specificity and affinity.	[7,8]
1997	First large-scale nanobody production and screening platform	The feasibility of producing and screening nanobodies from camelid heavy-chain antibodies was demonstrated, enabling their large-scale application, providing a robust platform for developing nanobodies with high specificity and stability.	[9]
2000s	Early applications in diagnostics	Nanobodies started being applied for diagnostic purposes, including biosensors and immunoassays. Their small size and stability made them suitable for rapid diagnostics, including the detection of bacteria, viruses, and toxins.	[10,11,12]
2001	The term “nanobody” was introduced	The term “nanobody” introduced by Ablynx in 2001 describes single-domain antibody fragments derived from camelid heavy-chain-only antibodies and is a registered trademark of Ablynx NV.	[8]
2005–2009	Humanization of nanobodies	Efforts to humanize nanobodies began, making them suitable for use in human therapies. This reduced their potential immunogenicity while preserving their functional activity.	[4,13]
2008	Nanobody for in vivo imaging	Nanobodies were successfully used for in vivo imaging due to their small size and ability to target tumor biomarkers, offering a more efficient alternative to traditional antibodies for molecular imaging.	[14,15,16]
2010s	Nanobody engineering advances	New techniques such as ribosome display and yeast display were employed to create large nanobody libraries, improving screening and selection processes for targeted nanobody therapies and diagnostics.	[17,18]
2018	First nanobody-based therapeutic approved	Caplacizumab (Cablivi), developed by Ablynx, became the first nanobody-based therapeutic approved in the EU for treating thrombotic thrombocytopenic purpura (TTP).	[19]
2021	First pan-tumor PD-L1 antibody and subcutaneous immunotherapy approved	Envafolimab (KN035, Envorria), co-developed by CanSino Biologics and 3D Medicines, became the first approved subcutaneous PD-L1 antibody and the first pan-tumor oncology therapy for unresectable or metastatic microsatellite instability-high/mismatch repair-deficient (MSI-H/dMMR) solid tumors in China.	[20]
2022	First nanobody-based CAR-T therapy approved	Carvykti (ciltacabtagene autoleucel), developed by Legend Biotech and Janssen, became the first CAR-T therapy using nanobody technology approved in the U.S. for relapsed/refractory multiple myeloma (RRMM). Its design incorporates a dual-BCMA-targeting nanobody.	[21]
2022	First nanobody for autoimmune disease approved	Ozoralizumab (Nanozora), developed by Ablynx, became the first nanobody-based therapy for autoimmune disease approved in Japan for treating rheumatoid arthritis (RA). Its trivalent bispecific VHH design combines two anti-TNFα domains (targeting inflammation) and one anti-HSA domain (prolonging half-life).	[22]
2020s	AI-driven nanobody design	The integration of AI in nanobody engineering began, with AI tools such as AlphaFold2 and ProteinMPNN enhancing the prediction of nanobody structures and enabling the rational design of multiepitope nanobodies for targeted therapeutic and diagnostic applications.	[6,23]

**Table 2 biology-14-00547-t002:** Comparison of the display technologies.

Display Technology	Description	Key Features	Advantages	Disadvantages
Phage Display [42]	Involves the expression of peptides or antibodies on the surface of bacteriophages, where the genetic information is linked to the displayed protein.	High-diversity libraries, easy cloning, rapid screening	High-throughput screening, easy to scale up, well established	Limited to larger targets, less efficient for membrane-bound proteins, phage amplification may be required
Ribosome Display [17,43]	An in vitro method where mRNA is linked to its translated protein, forming a stable mRNA–protein complex that can be used for screening.	No need for host cell transformation, no bacterial growth	No transformation required, high-diversity libraries (up to 10^13^), high sensitivity	Labor-intensive, slower screening process compared to phage display
Yeast Display [18]	Displays peptides or antibodies on the surface of yeast cells. The displayed proteins are directly linked to the yeast genome.	Yeast cells serve as both expression and selection systems	Single-cell resolution, real-time binding analysis, simpler than mammalian systems	Lower throughput compared to phage display, limited to eukaryotic targets, needs yeast transformation
Bacterial Display (Escherichia coli) [44,45]	Proteins are displayed on the surface of bacteria, which can then be selected based on binding to a target.	Fast expression, uses bacterial systems.	High expression levels, simple and cost-effective	Limited to smaller targets, lower display efficiency compared to yeast, less versatile for complex proteins
Mammalian Display [46]	A type of display where proteins are expressed on the surface of mammalian cells. This allows for the display of complex proteins.	Suitable for complex and membrane proteins, similar to human systems.	Better mimic of natural systems, good for membrane proteins and intracellular interactions	Expensive, requires specialized equipment, lower throughput than yeast or phage display
mRNA Display [47,48]	In vitro method where mRNA is linked to its corresponding protein, creating a stable mRNA–protein complex for screening.	Does not require cell-based transformation, used for in vitro selections	Huge diversity of libraries, high sensitivity and speed, can work without transformation	Requires specialized equipment, labor-intensive for large libraries

**Table 3 biology-14-00547-t003:** Summary of structural differences between camelid, human and shark derived nanobodies.

Property	Camelid-Derived (VHH)	Human-Derived	Shark-Derived (V_NAR_)
CDR3 Loop	Long and flexible	Shorter and less flexible	Elongated, highly flexible
Stability	High, due to hydrophilic framework mutations	Moderate, requires engineering	Extremely high, naturally stable
Size	~12–15 kDa	~15 kDa	~12 kDa (smallest fragment)
Antigen Access	Cryptic and concave epitopes	Protein antigens, less cryptic sites	Narrow grooves, cryptic epitopes
Natural Solubility	High	Moderate, engineered for solubility	Very high
Primary Application	Protein antigens (therapeutics, diagnostics)	Humanized therapies, low immunogenicity	Extreme environments, non-protein antigens

**Table 4 biology-14-00547-t004:** Overview of computational tools applicable to VHH or antibody structural modeling.

Model Name	VHH Applicability	Model Type	Main Function	Code Availability (Most Recent)	Web Server or Colab Availability (Most Recent)	Reference
AlphaFold2	Yes	Transformer	Protein monomer structure computation	https://github.com/google-deepmind/alphafold, accessed on 4 April 2025	https://colab.research.google.com/github/sokrypton/ColabFold/blob/main/AlphaFold2.ipynb, accessed on 4 April 2025	[23]
DeepAb	Yes	RNN(biLSTM + LSTM) ResNet	Antibody Fv and VHH structure prediction	https://github.com/RosettaCommons/DeepAb, accessed on 4 April 2025	https://colab.research.google.com/github/RosettaCommons/DeepAb/blob/main/DeepAb.ipynb, accessed on 4 April 2025	[144]
ABlooper	No	GNN(E(n)-EGNN)	Antibody CDR loop prediction	https://github.com/brennanaba/ABlooper, accessed on 4 April 2025	Not found	[145]
NanoNet	Yes	ResNet	High-throughput VHH structure determination	https://github.com/dina-lab3D/NanoNet, accessed on 4 April 2025	https://bio3d.cs.huji.ac.il/nanonet/, accessed on 4 April 2025	[149]
IgFold	Yes	Transformer(AntiBERTy [154] based on BERT)	Antibody and VHH structure prediction	https://github.com/Graylab/IgFold, accessed on 4 April 2025	https://colab.research.google.com/github/Graylab/IgFold/blob/main/IgFold.ipynb, accessed on 4 April 2025	[155]
ImmuneBuilder	Yes	Transformer(Based on AlphaFold-Multimer [171])	Antibody, VHH and T-cell receptor structure prediction	https://github.com/brennanaba/ImmuneBuilder, accessed on 4 April 2025	https://colab.research.google.com/github/brennanaba/ImuneBuilder/blob/main/notebook/ImmuneBuilder.ipynb, accessed on 4 April 2025	[156]
Parapred	No	RNN(LSTM) CNN	Antibody paratope prediction	https://github.com/eliberis/parapred, accessed on 4 April 2025	Not found	[157]
ParaAntiProt	Yes	Transformer(Based on ProtTrans [172]) CNN	Antibody and VHH paratope prediction	https://github.com/Alirzeanoroozi/ParaAntiProt, accessed on 4 April 2025	Not found	[158]
NbX	Yes	Decision Tree	Nanobody binding pose prediction	https://github.com/johnnytam100/NbX, accessed on 4 April 2025	Not found	[164]
DLAB	No	CNN	Antibody virtual screening	https://github.com/con-schneider/dlab-public, accessed on 4 April 2025	Not found	[165]
AlphaFold3	Yes	Transformer(Adapted from AF2 [23])	Structural modeling of monomers and multimers for biomacromolecules	https://github.com/google-deepmind/alphafold3, accessed on 4 April 2025	https://alphafoldserver.com/, accessed on 4 April 2025	[5]

## Data Availability

The datasets generated and analyzed during the current study are available from the corresponding author on reasonable request.

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
