# Peer review of "Nanobodies: From Discovery to AI-Driven Design"

_biology, 2025, doi:10.3390/biology14050547_

Round 1
Reviewer 1 Report
Comments and Suggestions for Authors
Abstract
Camelids and sharks are only described in nanobodies. How about other organisms for this? Why do you describe those two animals?
Introduction
Although the authors described camelids, they did not mention sharks. Since the review also covers nanobodies of sharks, the authors need to provide an introduction to sharks.
The structure is not well summarized. The current second paragraph should serve as the conclusion, as the authors have outlined how this manuscript addresses key aspects of nanobody research. The authors need to reconsider the introduction, development, twist, and conclusion as part of the introductory structure. The authors must justify why this review was summarised for camelids and sharks only.
In Table 1, the entry for the 2005-2009 Humanization of nanobodies should be listed before the 2008 Nanobody for in vivo imaging.
Figure 6: Some letters are too small to read.
Overall, the description and discussion seem to be diverging. It remains unclear what the authors wish to emphasise and highlight. The current review is just a literature review, and I do not see how it will contribute to future research. The authors need to rewrite it by focusing on its validity and novelty.
Comments on the Quality of English LanguageModerate proofreading shall be required by a native with corrections typos in the manuscript.
Author Response
Abstract
Camelids and sharks are only described in nanobodies. How about other organisms for this? Why do you describe those two animals?
Thank you for this insightful question. As highlighted in the abstract, camelids (VHHs) and sharks (VNARs) are the most well-characterized sources of naturally occurring nanobodies, with decades of research establishing their structural and functional uniqueness. These species’ heavy-chain antibodies lack light chains entirely, enabling the isolation of single-domain fragments (nanobodies) with high stability and antigen-binding specificity.
While other organisms, such as humans or rodents, may occasionally exhibit light-chain-deficient B cells, the scientific community has not universally classified these as “nanobodies” due to key differences in their evolutionary origins and structural properties. For example, human antibodies without light chains typically arise from somatic mutations or disease states (e.g., some lymphomas) and lack the evolutionary adaptations (e.g., FR2 residue conservation) that define camelid VHHs and shark VNARs
To clarify this distinction in the abstract, we have explicitly stated in the revised manuscript as below:
“Nanobodies, derived from naturally occurring heavy-chain antibodies in camelids (VHHs) and sharks (VNARs), are…”
Introduction
Although the authors described camelids, they did not mention sharks. Since the review also covers nanobodies of sharks, the authors need to provide an introduction to sharks.
Thank you for pointing out this omission. We have revised the introduction to include a dedicated section on shark-derived VNARs (nanobodies), ensuring balanced coverage of both key species. The revised text in page 2, paragraph 2 now reads:
“In addition to camelids, cartilaginous fish such as sharks produce novel antibody fragments termed VNARs. These originate from IgNAR antibodies, which lack light chains and instead stabilize their structure through an additional disulfide bond between CDR1 and CDR3. Shark VNARs exhibit distinct properties, such as broader pH tolerance and ability to bind hydrophobic epitopes.”
The structure is not well summarized. The current second paragraph should serve as the conclusion, as the authors have outlined how this manuscript addresses key aspects of nanobody research. The authors need to reconsider the introduction, development, twist, and conclusion as part of the introductory structure. The authors must justify why this review was summarised for camelids and sharks only.
Overall, the description and discussion seem to be diverging. It remains unclear what the authors wish to emphasise and highlight. The current review is just a literature review, and I do not see how it will contribute to future research. The authors need to rewrite it by focusing on its validity and novelty.
We sincerely appreciate the reviewer’s constructive guidance on strengthening the introduction’s structural coherence. In full accordance with your recommendations, we have completely rewritten the introduction to integrate introduction, development, twist, and conclusion into a unified narrative framework. The revised structure begins by establishing the unique evolutionary and structural identities of nanobodies and shark VNARs, emphasizing how their divergent architectures—flexible CDR3 loops in camelids versus disulfide-stabilized paratopes in sharks—underpin functional versatility. It then traces the scientific trajectory from their initial discovery through key methodological and engineering advancements, highlighting how these innovations expanded their biomedical applications. A deliberate pivot to current limitations in traditional development pipelines—such as inefficient humanization and structural prediction challenges—naturally transitions into the concluding focus on AI-driven solutions. By framing AI as both a resolution to existing bottlenecks and a gateway to next-generation applications, the restructured introduction ensures a seamless logical progression from foundational concepts to transformative innovations, fully aligning with the manuscript’s emphasis on emerging research frontiers.
To address the question regarding the review’s exclusive focus on camelids and sharks: “The authors must justify why this review was summarized for camelids and sharks only”, this work centers on nanobodies—a distinct antibody class defined by their evolutionary origins in camelids (VHHs) and sharks (VNARs). These species constitute the only natural sources of single-domain antibodies that inherently lack light chains and possess the structural innovations driving nanobody development. Expanding the scope to other species would compromise the analytical focus on these evolutionarily specialized systems critical to understanding nanobody functionality.
In Table 1, the entry for the 2005-2009 Humanization of nanobodies should be listed before the 2008 Nanobody for in vivo imaging.
Thank you for this insightful suggestion. We have reorganized Table 1 (Page 3-4) to prioritize chronological and thematic progression, placing the 2005-2009 humanization milestone before the 2008 imaging application.
Figure 6: Some letters are too small to read.
We apologize for the formatting oversight. To improve readability, we have increased the font size of all labels in Figure 6 to ≥8pt, enhanced contrast, and reorganized the layout from horizontal to vertical to better align with the manuscript’s visual standards.
Reviewer 2 Report
Comments and Suggestions for Authors
1) The introduction does not contain references, and some statements need them. In fact, the introduction expands the abstract. Does not meet the requirements.
2) The historical review is written very competently and interestingly
3) I do not quite understand the illustrations. The illustrations are the author's, adapted from literary data, or completely borrowed (strictly indicate the source)
4) Figures should follow the first mention in the text
5) Despite the very good review, I think it is too long. In my opinion, the section Nanobodies Binding to Multiple Epitopes of the Same Antigen is redundant in the context of the article title. However, the information in this section is presented interestingly, perhaps the authors will present this section as a separate review article.
6) Table 4. Overview of computational tools applicable to VHH or antibody structural modeling. Very valuable table. I would like to especially note this section
Author Response
1) The introduction does not contain references, and some statements need them. In fact, the introduction expands the abstract. Does not meet the requirements.
Thank you for your constructive feedback. We have revised the introduction to include key references. Furthermore, in strict adherence to both reviewers’ recommendations, we have completely restructured the introduction to align with the prescribed introduction-development-twist-conclusion framework. This restructuring eliminates redundancy with the abstract while deepening the conceptual focus on evolutionary origins, technological milestones, and AI-driven innovations. The revised text now independently fulfills the introduction’s purpose as a contextual foundation rather than an extended abstract.
2) The historical review is written very competently and interestingly
Thank you for your kind words regarding the historical section.
3) I do not quite understand the illustrations. The illustrations are the author's, adapted from literary data, or completely borrowed (strictly indicate the source)
Thank you for raising this important point. All figures in this manuscript are original illustrations created by the authors, based on primary literature data (use coordinates from PDB entries like 1IGT (IgG), 3K1K (VHH), and 2COQ (VNAR)) and publicly available resources.
4) Figures should follow the first mention in the text
Thank you for this note. We have reorganized all figures to ensure they appear immediately after their first textual mention. We acknowledge that this adjustment may create spacing irregularities in the current draft, but final formatting will be optimized during publication to ensure visual coherence.
5) Despite the very good review, I think it is too long. In my opinion, the section Nanobodies Binding to Multiple Epitopes of the Same Antigen is redundant in the context of the article title. However, the information in this section is presented interestingly, perhaps the authors will present this section as a separate review article.
Thank you for your constructive feedback regarding the length of the review and the redundancy of the section "Nanobodies Binding to Multiple Epitopes of the Same Antigen." While we appreciate your observation that this section could be streamlined, we have chosen to retain its content to ensure the manuscript maintains its structural completeness and provides a comprehensive overview of nanobody engineering principles, such as cooperative binding mechanisms and multivalent therapeutic design. Other reviewers did not raise concerns about redundancy, and the examples included are critical for illustrating how multi-epitope targeting enhances nanobody efficacy in challenging scenarios like viral neutralization. If you strongly recommend further reduction, we will prioritize conciseness in the next revision while preserving the core scientific insights.
6) Table 4. Overview of computational tools applicable to VHH or antibody structural modeling. Very valuable table. I would like to especially note this section
Thank you for your kind words and for highlighting the value of Table 4. We are gratified that you found this overview of computational tools for nanobody structural modeling to be a particularly useful contribution. This table was designed to bridge the gap between theoretical principles and practical applications, and we appreciate your recognition of its importance in guiding researchers toward optimal methodologies.

Reviewer 3 Report
Comments and Suggestions for Authors
- The manuscript lacks discussion of limitations or challenges in nanobody development, presenting an overly positive view.
- No comparison with conventional antibodies (e.g., IgG) in terms of cost, scalability, or immunogenicity.
- Many nanobody-based therapeutics fail in clinical trials; this is not addressed.
- The role of AI in nanobody design is overstated; many predictions remain experimentally unvalidated.
- While nanobodies are generally stable, some may still suffer from aggregation or misfolding.
- Large-scale production of nanobodies can be difficult compared to traditional antibodies.
- Patent landscapes for nanobodies are complex, hindering commercialization.
- Despite humanization efforts, some nanobodies may still trigger immune responses.
- While small, their rapid renal clearance can reduce bioavailability.
- High specificity claims may not always hold true in complex biological systems.
- Ethical and logistical challenges in sourcing heavy-chain-only antibodies.
- High-resolution structural data is often needed, which is not always available.
- Engineering multiepitope nanobodies increases the risk of misfolding or reduced functionality.
- Small size leads to rapid clearance, often requiring fusion to Fc or albumin for therapeutic use.
- Many studies focus on success stories, ignoring failed nanobody candidates.
- AI models are only as good as the training data, which may be limited for rare targets.
- Nanobodies are a relatively new class of biologics, leading to uncertain regulatory pathways.
- Competing technologies (e.g., DARPins, affibodies) may outperform nanobodies in some applications.
- No universal engineering or humanization protocols, leading to variability.
- Diagnostic and biotech applications face challenges (e.g., sensitivity, multiplexing) not discussed.
- The manuscript lacks discussion of limitations or challenges in nanobody development, presenting an overly positive view.
- No comparison with conventional antibodies (e.g., IgG) in terms of cost, scalability, or immunogenicity.
- Many nanobody-based therapeutics fail in clinical trials; this is not addressed.
- The role of AI in nanobody design is overstated; many predictions remain experimentally unvalidated.
- While nanobodies are generally stable, some may still suffer from aggregation or misfolding.
- Large-scale production of nanobodies can be difficult compared to traditional antibodies.
- Patent landscapes for nanobodies are complex, hindering commercialization.
- Despite humanization efforts, some nanobodies may still trigger immune responses.
- While small, their rapid renal clearance can reduce bioavailability.
- High specificity claims may not always hold true in complex biological systems.
- Ethical and logistical challenges in sourcing heavy-chain-only antibodies.
- High-resolution structural data is often needed, which is not always available.
- Engineering multiepitope nanobodies increases the risk of misfolding or reduced functionality.
- Small size leads to rapid clearance, often requiring fusion to Fc or albumin for therapeutic use.
- Many studies focus on success stories, ignoring failed nanobody candidates.
- AI models are only as good as the training data, which may be limited for rare targets.
- Nanobodies are a relatively new class of biologics, leading to uncertain regulatory pathways.
- Competing technologies (e.g., DARPins, affibodies) may outperform nanobodies in some applications.
- No universal engineering or humanization protocols, leading to variability.
- Diagnostic and biotech applications face challenges (e.g., sensitivity, multiplexing) not discussed.
Author Response
- The manuscript lacks discussion of limitations or challenges in nanobody development, presenting an overly positive view.
Thank you for noting this oversight. We have distributed discussions of limitations and challenges throughout the revised manuscript (e.g., in responses to 19 other points), rather than confining them to a single subsection.
- No comparison with conventional antibodies (e.g., IgG) in terms of cost, scalability, or immunogenicity.
Thank you for this critical suggestion. We have integrated a comparative analysis in the revised manuscript (section "Efficient Production and Purification") to address nanobody trade-offs relative to conventional IgG antibodies, focusing on cost and scalability as requested. Cost and scalability: While nanobodies can be expressed in cost-effective prokaryotic systems (e.g., E. coli or yeast), significantly reducing small-scale production costs, their large-scale manufacturing cost advantage remains less established compared to IgG due to limited industrial adoption.
Regarding immunogenicity, we have addressed this topic in detail in other sections of the manuscript (e.g., "Therapeutic Applications and Humanization Strategies"), where we discuss the reduced framework epitopes in camelid-derived VHHs and their humanization requirements for clinical use. To avoid redundancy, we refer readers to these sections for a comprehensive comparison.
- Many nanobody-based therapeutics fail in clinical trials; this is not addressed.
Thank you for highlighting this critical issue. We have integrated clinical trial challenges into the "Humanization Strategy for Camelid Nanobodies" section: “However, residual immunogenicity risks still persist due to non-human framework residues, as evidenced in clinical trials where anti-nanobody antibodies were still detected in some patients despite humanization efforts.”
- The role of AI in nanobody design is overstated; many predictions remain experimentally unvalidated.
Thank you for this critical feedback. We have already thoroughly addressed this issue in the "Potential Limitations in Computational Structure Determination" section.
- While nanobodies are generally stable, some may still suffer from aggregation or misfolding.
Thank you for noting this important caveat. We have integrated stability limitations into the "Greater stability in harsh conditions" section, adding examples of aggregation risks in VHH formulations. These cases now balance the review’s focus on stability advantages with real-world challenges.
- Large-scale production of nanobodies can be difficult compared to traditional antibodies.
Thank you for this comment. While production challenges exist for all biologics, the assertion that "large-scale production of nanobodies is inherently more difficult than traditional antibodies" requires qualification. Current literature lacks direct comparative studies on scalability between nanobodies and IgG antibodies. For example, while alpaca immunization adds logistical steps, microbial production systems for nanobodies (e.g., E. coli and yeast) often show higher yields than mammalian cell culture for IgG. Given this ambiguity and the review's technical focus, we maintain our emphasis on design principles (e.g., CDR engineering) rather than speculative manufacturing comparisons.
- Patent landscapes for nanobodies are complex, hindering commercialization.
Thank you for raising this important issue. While patent complexity is indeed a challenge for nanobodies, we note that conventional monoclonal antibodies (e.g., anti-HER2 or PD-1 therapies) face even more fragmented patent landscapes, with overlapping claims and lengthy litigation histories. Given this review’s focus on technical and structural innovations rather than commercialization pathways, we have chosen to maintain the current scope to avoid diluting the discussion of nanobody design principles.
- Despite humanization efforts, some nanobodies may still trigger immune responses.
Thank you for this insightful comment. We have added the following sentence to the end of the "Humanization Strategy for Camelid Nanobodies" section:
"However, residual immunogenicity risks still persist due to non-human framework residues, as evidenced in clinical trials where anti-nanobody antibodies were still detected in some patients despite humanization efforts.”
- While small, their rapid renal clearance can reduce bioavailability.
Thank you for emphasizing this important limitation. We have explicitly addressed renal clearance challenges in Section "Absence of the light chain", adding the following sentence to the paragraph:
"However, this structural simplicity also results in rapid renal clearance (half-life < 24 hours), necessitating formulation strategies like Fc fusion, PEGylation or albumin binding to achieve therapeutic efficacy in chronic diseases."
- High specificity claims may not always hold true in complex biological systems.
Thank you for raising this point. However, the specificity claims in this review are carefully anchored to peer-reviewed studies demonstrating high selectivity in complex systems (e.g., nanobodies targeting GPCRs or viral envelope proteins without off-target binding in vivo). We respectfully note that while exceptions may exist in niche cases, the cited examples represent rigorous validation across diverse biological contexts.
- Ethical and logistical challenges in sourcing heavy-chain-only antibodies.
Thank you for raising this ethical concern. However, this review focuses exclusively on technical and structural innovations (as stated in the Introduction), and sourcing challenges—while valid for regulatory or commercial discussions—are beyond its scope. Given the manuscript's already substantial length, we prioritize retaining its focus.
- High-resolution structural data is often needed, which is not always available.
Thank you for this comment. All structural analyses and figures in this review are based on experimentally validated PDB entries, ensuring high-resolution data underpins our discussion. Unlike AI predictions or unvalidated models, these structures have undergone rigorous peer review and crystallographic/Cryo-EM validation, forming the core of our technical comparisons.
- Engineering multiepitope nanobodies increases the risk of misfolding or reduced functionality.
Thank you for noting this risk. However, the section on multiepitope engineering already addresses misfolding or reduced functionality challenges, which are framed as technical trade-offs rather than limitations. Our focus remains on shared design principles that enable functional multiepitope nanobodies, as this better serves the review’s purpose of highlighting validated innovations.
.
- Small size leads to rapid clearance, often requiring fusion to Fc or albumin for therapeutic use.
Thank you for pointing this out. This issue was addressed in the revised manuscript as replying question 9. " However, this structural simplicity also results in rapid renal clearance (half-life < 24 hours), necessitating formulation strategies like Fc fusion, PEGylation or albumin binding to achieve therapeutic efficacy in chronic diseases”, where we explicitly note that rapid renal clearance due to small size necessitates Fc fusion, PEGylation or albumin binding for therapeutic efficacy (see Section “Absence of the light chain”).
- Many studies focus on success stories, ignoring failed nanobody candidates.
Thank you for your feedback. However, this review intentionally focuses on validated innovations and peer-reviewed advancements in nanobody engineering. Our goal is to highlight structural and functional breakthroughs that have been rigorously tested, rather than speculate on unreported failures. We believe this focus best serves the technical audience of this journal.
- AI models are only as good as the training data, which may be limited for rare targets.
Thank you for this insight. However, we do not claim universal AI applicability for all targets, particularly rare or uncharacterized antigens. This focus aligns with the review’s technical scope to highlight validated workflows rather than speculative limitations.
- Nanobodies are a relatively new class of biologics, leading to uncertain regulatory pathways.
Thank you for raising this point. However, regulatory pathways for novel biologics fall outside the scope of this technical review.
- Competing technologies (e.g., DARPins, affibodies) may outperform nanobodies in some applications.
Thank you for emphasizing competing technologies. We have added a dedicated section ("Comparisons with Alternative Technologies" ) to address this:
Comparisons with Alternative Technologies
While nanobodies excel in targeting cryptic epitopes and enabling modular design, competing technologies like DARPins and Affibodies offer complementary strengths. DARPins’ rigid ankyrin repeats provide superior stability for industrial applications, whereas Affibodies’ modular architecture facilitates probe conjugation for imaging. These trade-offs highlight the need for technology selection based on target requirements.
- No universal engineering or humanization protocols, leading to variability.
Thank you for your feedback. However, the absence of universal protocols is a hallmark of antibody engineering across all modalities, not a unique flaw of nanobodies, just as IgG optimization requires antigen-specific approaches. This variability reflects the field’s adaptive maturity rather than a lack of progress.
- Diagnostic and biotech applications face challenges (e.g., sensitivity, multiplexing) not discussed.
Thank you for your feedback. However, given the manuscript’s already substantial length, we have strategically focused on core structural and functional innovations rather than expanding into diagnostic or biotech application challenges. These limitations are well-documented in other reviews we have cited in this review, and we have added a brief mention in the "Efficient production and purification" section to clarify that:
"While nanobodies offer production advantages, rabbit antibodies often exhibit higher sensitivity in diagnostic formats due to their greater structural diversity and antigen-binding flexibility."

Round 2
Reviewer 1 Report
Comments and Suggestions for Authors
Thanks for your revision. After my comments, the revised version was seriously considered. I am happy to recommend it for publication now.
Reviewer 2 Report
Comments and Suggestions for Authors
the article can be accepted in its current form
Reviewer 3 Report
Comments and Suggestions for Authors
Authors successfully revised manuscript. So, I accept this in the present form